



# Vectors to ore in replacive VMS deposits of the northern Iberian Pyrite Belt: mineral zoning, whole rock geochemistry, and use of portable XRF

Guillem Gisbert[1], Fernando Tornos[1], Emma Losantos[1], Juan Manuel Pons[2], Juan Carlos Videira[2]

[1] Institute of Geosciences (CSIC-UCM), Madrid, 28040, Spain
[2] MATSA, Almonaster la Real, Huelva, 21342, Spain

*Correspondence to*: Guillem Gisbert (ggisbertp@hotmail.com)

**Abstract.** Volcanogenic Massive Sulphide (VMS) deposits represent a major source of base, precious and other metals of economic and industrial importance. As in other mineral systems, progressive exhaustion of the shallowest and most easily

accessible deposits is leading to increasingly complex exploration. In this context vectors to ore play a vital role. The Iberian Pyrite Belt (IPB) is an outstanding VMS district located in the SW Iberian Peninsula, which represents the main mining area in Spain and one of the main zones of base metal production in Europe. But the work on vectors to ore in the IPB is far from systematic or complete.

In this work we have performed a detailed study of the main vectors to ore related to mineral zoning and whole rock

geochemistry that are currently used in the exploration of VMS systems to a representative volcanic rock hosted replacive VMS deposit located in the northern IPB, the Aguas Teñidas deposit. Results have been compared to other deposits in the IPB and in other VMS districts. The investigated vectors include: mineralogical zoning, host sequence characterization and mineralized unit identification based on whole rock geochemistry, the study of the characteristics and behaviour of whole rock geochemical anomalies around the ore (e.g. alteration-related compositional changes, characteristics and extent of

geochemical halos around the deposit), with definition of threshold values for the mineralization-related indicative elements, and application of portable XRF analysis to the detection of the previous vectors.

In the footwall, a concentric cone-shaped hydrothermal alteration bearing the stockwork passes laterally, from core to edge, from quartz (only locally), to chlorite, sericite–chlorite, and sericite alteration zones. The hydrothermal alteration is also found in the hanging wall despite its thrusted character: a proximal sericite alteration zone is followed by a more distal albite

one, which is described here for the first time in the IPB. Whole rock major elements show an increase in alteration indexes (e.g. AI, CCPI) towards the mineralization, with a general $SiO_2$ enrichment, FeO enrichment in the central portion of the system, $K_2O$ and $Na_2O$ leaching towards the outside areas, and a less systematic MgO behaviour. Copper, Pb and Zn produce proximal anomalies around mineralized areas, with the more mobile Sb, Tl and Ba generating wider halos. Whereas Sb and Tl halos form around all mineralized areas, Ba anomalies are restricted to areas around the massive sulphide body. Our

results show that proposed vectors, or adaptations designed to overcome p-XRF limitations, can be confidently used by analysing unprepared hand specimens, including the external rough curved surface of drill cores.

The data presented in this work are not only applicable to VMS exploration in the IPB, but on a broader scale they will also contribute to improve our general understating of vectors to ore in replacive-type VMS deposits.

## 1 Introduction

Volcanogenic Massive Sulphide (VMS) deposits represent a major source of base (Cu, Pb, Zn), precious (Ag, Au) and other metals (e.g. Co, Sn, In, Cd, Tl, Ga, Se, Sb, Bi) of economic and industrial importance (Large et al., 2001a; Franklin et al., 2005; Tornos et al., 2015). They are distributed in discrete provinces worldwide (e.g. Iberian Pyrite Belt, Spain, Tornos et al., 2006; Bathurst Mining Camp, Canada, Goodfellow and McCutcheon, 2003; Mount Read, Australia, Large et al., 2001a;



Kuroko, Japan, Ohmoto, 1996). With progressive exhaustion of the shallowest and most easily accessible ore deposits,
exploration for new resources faces challenges such as exploration at increasing depths, under covering (e.g. unrelated lithological or tectonic units, urbanized areas) or in non-conventional settings, and an inevitable need for improved efficiency and lower impact, both in environmental and social terms. In this context, the combined study of the mineral systems and the development of new exploration strategies and technologies based on geophysical methods and vectors to ore play a vital role.


The use of vectors to ore focuses on the identification and study of lithogeochemical fingerprints produced by the mineralizing hydrothermal system or by subsequent ore remobilizations within and around ore deposits (e.g. Madeisky and Stanley, 1993; Large et al., 2001a; Ames et al., 2016). Vectors to ore have the potential to detect the nearby presence of an ore deposit and to provide information on its likely location or characteristics. They are typically based on the observation of
variations in lithology, geochemistry, mineralogy, and mineral chemistry (e.g. Ballantyne, 1981; Large et al., 2001b; Cooke et al., 2017; Mukherjee and Large, 2017; Soltani Dehnavi et al., 2018a, Hollis et al., 2021),  and are characteristic to each deposit type (e.g. trace element mineral chemistry in porphyry-Cu systems: Cooke et al., 2014, 2017; in VMS systems: Soltani-Dehnavi et al., 2018a, b, 2019; in SEDEX systems: Mukherjee and Large, 2017). Additionally, their behaviour may change from district to district, which makes specific characterization of vectors in each district a necessary task for their
correct use. Main vectors to ore currently used in VMS systems are shown in Figure 1.

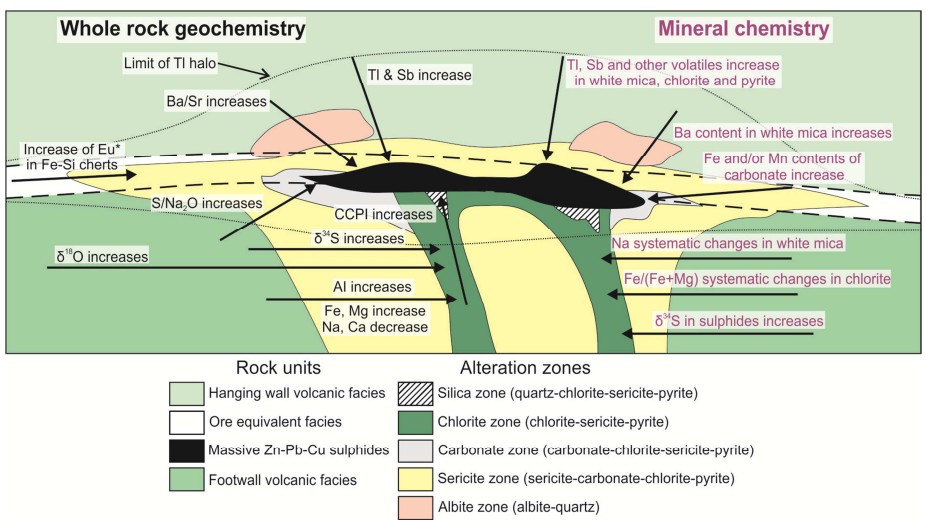

**Figure 1: Mineralogical and geochemical halos around VMS deposits. Modified from Large et al. (2001a) with data from Lydon (1988), Cathles (1993), Gale and Fedikow (1993), Lenz and Goodfellow (1993), Goodfellow and Peter (1996), Lentz et al. (1997),**
**Brauhart et al. (2001), Large et al. (2001a, b), Gale (2003), Ames et al. (2016), and Soltani Dehnavi et al. (2018a,b, 2019).**

The Iberian Pyrite Belt (IPB) is an outstanding VMS district located in the SW Iberian Peninsula (Fig. 2). It is arguably the largest known accumulation of sulphides on the Earth's crust (Tornos, 2006), and represents the main mining area in Spain and one of the main zones of base metal production in Europe. The characterization of vectors to ore in the IPB (e.g. Relvas
et al., 1990; Madeisky and Stanley, 1993; Toscano et al., 1994; Costa, 1996; Relvas et al., 2006; Velasco-Acebes et al., 2019), is far from systematic or complete, especially when compared to the work done in other VMS districts (e.g. Australian districts, Large et al., 2001a and references therein; Canada, Soltani Dehnavi et al., 2018a, b; 2019). In addition, previous works have mostly focused on the study of the larger exhalative shale-hosted deposits of the southern IPB (e.g. Toscano et al., 1994; Tornos et al., 2008; Sáez et al., 2011; Velasc-Acebes et al., 2019) or the giant Rio Tinto deposit (e.g.



Madeisky and Stanley, 1993; Costa, 1996). However, less attention has been paid to the predominantly volcanic rock hosted
        replacive deposits of the northern IPB (e.g. Relvas, 1990; Sánchez-España et al., 2000), which, although generally smaller in
        size compared to southern deposits, typically present higher base metal concentrations (Tornos, 2006).

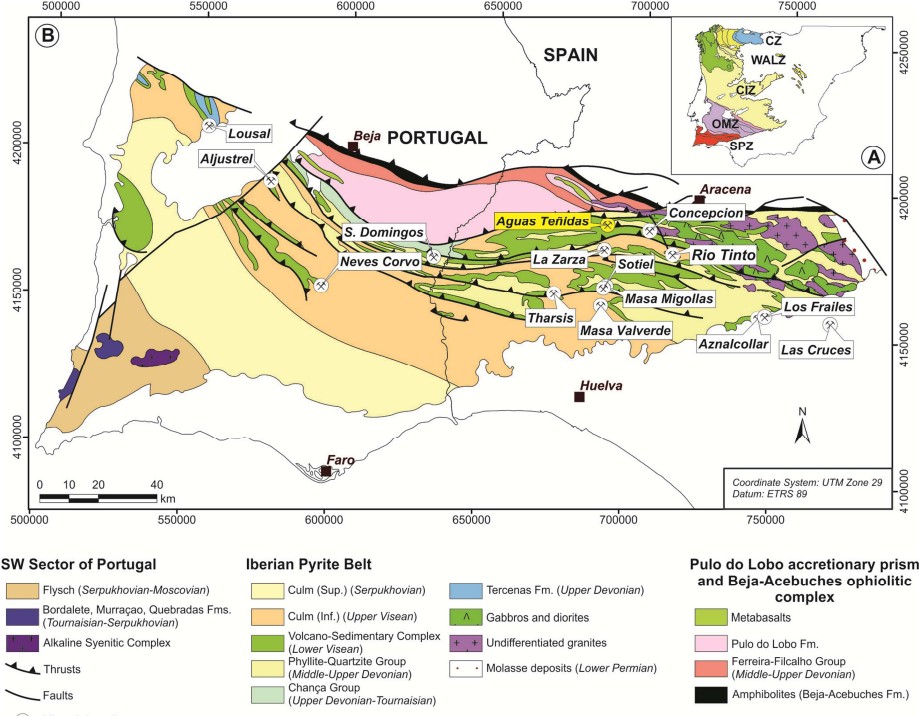

**Figure 2: Geological map of the South Portuguese Zone and location of Aguas Teñidas deposit. CZ: Cantabrian Zone; WALZ:**
        **West Asturian-Leonese Zone; GTOM: Galicia-Trás-os-Montes Zone: CIZ: Central Iberian Zone; OMZ: Ossa-Morena Zone;**
        **SPZ: South Portuguese Zone. Adapted from Martin-Izard et al. (2015), based on IGME (1982).**

        The aim of this work is to contribute to vector characterization in the IPB by focusing on the study of a representative
volcanic rock hosted replacive VMS deposit located in the northern IPB in order to improve mineral exploration and the
        location of new resources in the area. We have performed a thorough study of Aguas Teñidas deposit, on which the main
        vectors to ore currently used in the exploration of VMS systems have been investigated. Here we present the results of the
        study and characterization of vectors associated to mineralogical zoning and whole rock geochemistry. The latter have
        included identification of the mineralized unit, study of the characteristics and behaviour of whole rock geochemical
anomalies around the ore - with definition of mineralization-related threshold values -, and the assessment of the
        applicability of portable XRF analysis to their detection. New data are compared to data from previous studies on other
        deposits from this district as well as from other provinces. Data presented in this work will not only be applicable to
        exploration in the IPB, but on a broader scale will also contribute to improve our general understanding of vectors to ore in
        replacive-type VMS deposits.



## 2 Geological setting

### 2.1 Vectors to ore in VMS systems

Upwelling hot hydrothermal fluids in VMS systems react with the ambient rocks and sediments producing chemical and mineralogical changes in the form of alteration halos. These are most developed in the footwall, around the feeder zone, although they can also form, to a lesser extent, around the deposit and in its hanging wall, especially in sub-seafloor replacive deposits (Large et al., 2001a; Franklin et al., 2005; Hannington, 2014) (Fig. 1). The alteration assemblage is controlled by factors such as rock/sediment composition, water/rock ratio, temperature, and fluid composition, pH, and redox state (Hannington, 2014). Changes in these controlling factors with depth and distance from de centre of the hydrothermal system typically result in distinct mineralogically and geochemically zoned alteration halos, which are the base for most vectoring tools used in the exploration and characterization of VMS systems (Large et al. 2001a, Gibson et al., 2007) (Fig. 1). The size and morphology of the alteration zones is also controlled by differences in permeability and porosity of the host sequence (Large et al., 2001a, Gibson et al., 2007). However, it is important to bear in mind that alteration mineral assemblages and geochemical zonings are not only the result of the hydrothermal metasomatism associated with the mineralizing event. Instead, they are the result of the combined effects of the initial rock composition and all the processes that have modified it, such as seafloor metasomatism, hydrothermal metasomatism/s, metamorphism, and weathering (Madeisky and Stanley, 1993).

The main vectors to ore used in the exploration and study of VMS systems can be grouped into three main categories: 1) mineralogical zoning; 2) related to whole rock geochemistry; 3) related to mineral chemistry (Fig. 1). In addition, other tools such as identification of mineralization-related lithologies or lithologies indicative of favourable environmental conditions (e.g. anoxic stratigraphic horizons favourable for the formation and preservation of exhalative VMS deposits; Tornos et al., 2015) can be also considered as vectors to ore or pathfinders useful for mineral exploration.

### 2.2 VMS deposits in the Iberian Pyrite Belt

The IPB belongs to the southernmost domain of the Variscan Belt in the Iberian Peninsula: the South Portuguese Zone (Julivert et al., 1974) (Fig. 2). It holds over 1600 Mt of massive sulphides originally in place, and about 250 Mt of stockwork ore in over 90 VMS deposits (Tornos, 2006), including 22% of the VMS world class deposits (>32 Mt; Laznicka, 1999; Tornos, 2006). Individual massive sulphide bodies can be up to 170 Mt (La Zarza), but most giant deposits (e.g. Neves Corvo, Tharsis, Río Tinto) include 2 to 6 separate bodies located within an area of few square kilometres (Tornos, 2006).

The IPB VMS deposits were formed from Late Famennian to early Late Visean times within a transient transtensional intra-continental pull-apart basin generated on the South Portuguese Zone during the geodynamic evolution leading to the growth of the Variscan orogen in late Paleozoic (Oliveira, 1990). Crustal thinning and magmatic intrusion triggered the hydrothermal circulation responsible for the massive sulphide mineralizing events (Oliveira, 1990; Oliveira et al., 2004; Mitjavila et al., 1997; Tornos, 2006). A detailed review on the geology and mineralization processes of the IPB can be found in Barriga (1990), Leistel et al. (1998), Carvalho et al. (1999) and Tornos (2006) and is briefly summarized in Supplementary Material 1.1.

The stratigraphic sequence of the IPB records the pre-, syn-, and post-collisional evolution of the northern continental margin of South Portuguese Zone terrane. It consists of a 1000-5000 m thick (base not exposed; Tornos, 2006) Devonian to Carboniferous (Oliveira, 1990; Oliveira et al., 2004) sequence which has been divided into three main units (Schermerhorn, 1971), from older to younger: 1) Phyllite-Quartzite Group (PQ), 2) Volcanic Sedimentary Complex (VSC), which hosts the



mineralization, and 3) Culm Group or Baixo Alentejo Flysch Group. The host VSC consists of a complex bimodal volcanic and shallow intrusive sequence of mantle-derived mafic magmas and crustal felsic magmas, interbedded with mudstone and minor chemical sediments (Munhá, 1983; Mitjavila et al., 1997; Thiéblemont et al., 1998, Tornos, 2006). The VSC in the northern area of the IPB – where the Aguas Teñidas deposit is located - is dominated by volcanic materials with minor fine-
grained sediments with limited continental influence, whereas the southern area is dominated by shales and siliciclastic sediments with continental influence and minor volcanic and subvolcanic materials (Quesada, 1996; Sáez et al., 1999; Conde and Tornos, 2019). Two main contrasting styles of VMS mineralization have been described in the IPB which are closely related to the nature of the host stratigraphic sequence: dominantly exhalative shale-hosted deposits (e.g. Sotiel-Migollas, Tharsis, Lousal, Las Cruces, Aznalcóllar-Los Frailes, Masa Valverde) and dominantly replacive felsic-volcanic-rocks-hosted
deposits (e.g Aguas Teñidas, La Zarza, Aljustrel, Concepción, La Magdalena) (Relvas et al., 2001, 2002; Tornos et al., 1998, 2008; Tornos, 2006; Tornos and Heinrich, 2008; Velasco-Acebes et al. 2019).

Subsequent to deposition of the VSC, compressive tectonism lasted from Late Visean to Late Moscovian (Oliveira et al., 1979; Silva et al., 1990; Pereira et al., 2008). It disrupted the stratigraphic record of the IPB forming a S (Spain) and SW
(Portugal) -verging and -propagating thin-skinned foreland fold and thrust belt (Ribeiro and Silva, 1983; Silva et al., 1990; Quesada, 1991, 1998). The original geometry of VMS deposits was modified by tectonic dismembering and stacking during this stage (e.g. Relvas et al., 1990; Leistel et al., 1998; Quesada, 1998; Tornos et al., 1998). Associated to compressive tectonics, low-grade regional metamorphism, from prehnite-pumpellyite to low green schist facies, affected rocks in the IPB (Schermerhorn, 1975; Munhá, 1979, 1983, 1990, Sánchez España, 2000).
Numerous stratiform Fe-Mn ores within the VSC - mainly jaspers, which may occur laterally to massive sulphides- and non-economic late vein mineralization within the VSC and Culm Group - interpreted as produced by metal remobilization from the massive sulphides during the late stages of the Variscan orogeny- (Carvalho et al., 1999; McKee, 2003) occur in addition to the massive sulphides.

### 2.3 Aguas Teñidas deposit
Aguas Teñidas is a currently mined polymetallic (Cu-Zn-Pb) massive sulphide deposit located in the northern part of the IPB (Fig. 2b). It was discovered in 1985 during brownfield exploration in the area around the old Aguas Teñidas mine; further details on the discovery and mining history of this deposit are provided in Supplementary Material 1.2.

#### 2.3.1 Deposit characteristics
The massive sulphide body has no surface expression (Fig. 3, 4); it is elongated in a roughly E-W direction, with a plunge of around 20° to the west, at a depth between 280 m (eastern side) and 650 m (western side). It is wedge-shaped perpendicular to elongation, at least 1,800 m long, between 150 and 300 m wide, and with a maximum thickness of 90-100 m by its mostly fault-bounded northern margin (Hidalgo et al., 2000; McKee, 2003). It has an associated stockwork forming a discordant east-west trending funnel-shaped (in cross section) zone along the entire deposit (Bobrowicz, 1995; Hidalgo et al., 2000;
McKee, 2003) (Fig. 3, 4). The orebody is intensely deformed, with shear zones along several contacts with the host rock, and also within the main ore body (McKee, 2003). Whereas in most areas there is structural continuity between the massive sulphides and the underlying stockwork, the contact between the massive sulphides body and the hanging wall sequence usually consists of a shear zone of unknown displacement (Bobrowicz, 1995; Hidalgo et al., 2000; McKee, 2003). This is common in the IPB, where the hanging wall of the massive sulphides is typically thrusted over them (Tornos, 2006). The
preferential formation of thrusts above massive sulphides has been interpreted as a result of the rheological contrast between massive sulphides and host rocks (Quesada, 1998). The E-W Northern Fault at Aguas Teñidas has been interpreted as a syn-





sedimentary growth fault which acted as the feeder structure to the hydrothermal system (Bobrowicz, 1995; McKee, 2003), and which was reactivated during the Variscan compressive stage (McKee, 2003).

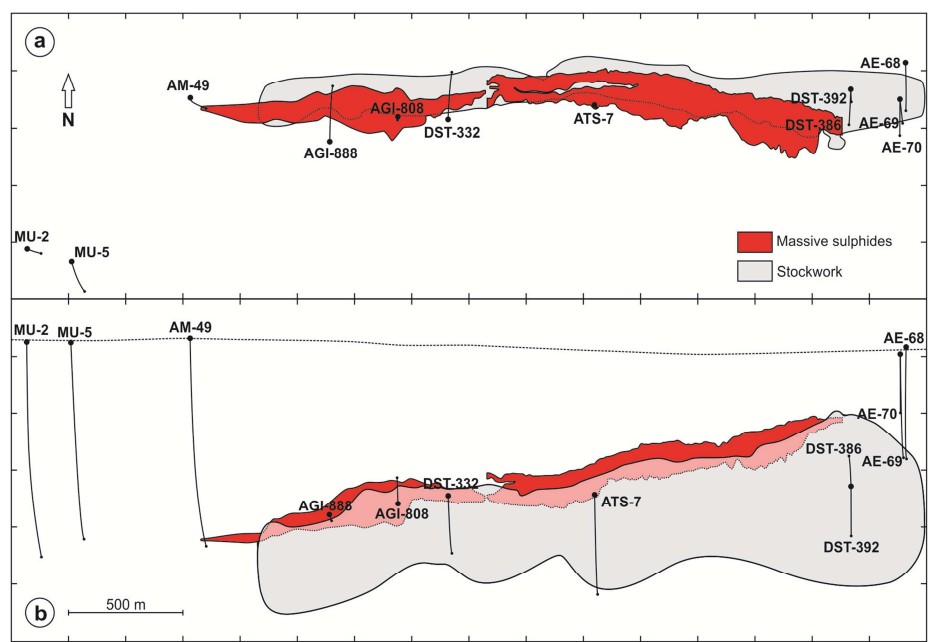


**Figure 3: Top (a) and front (b) view of the Aguas Teñidas deposit (massive sulphides and stockwork) and location of the studied drill cores. Dotted line in (b) represents ground level. Data provided by MATSA.**

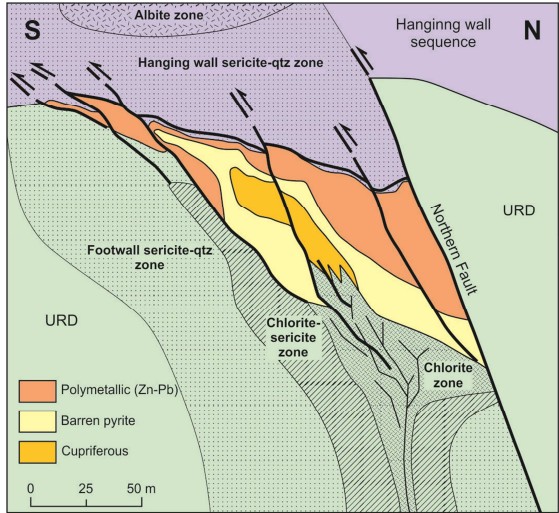

**Figure 4: Schematic cross section of Aguas Teñidas deposit based on España et al. (2003). It includes shear zones described by**
**McKee (2003) at the top of the massive sulphides and observed in this study, as well as hanging wall alteration zones described in this study.**

At a large, deposit, scale, the massive sulphide body has a mineral zonation similar to that in other VMS deposits, with a Cu-rich core at the base and Zn-Pb-rich ore towards the top and periphery, although a minor occurrence of Pb, Zn and Au at the
footwall contact deviates from the classical VMS model (Bobrowicz, 1995; McKee, 2003). At decimetre-scale it can be highly complex with many repetitions, displacements, and lateral variations (McKee, 2003).



Pyrite, sphalerite, chalcopyrite and galena account for over 95% of the massive sulphides, with pyrite generally constituting between 50 and 80% of the massive sulphide (Hidalgo et al. 2000). Tetrahedrite-tennantite group minerals, arsenopyrite,

stannite, bournonite and native bismuth are also present, as well as trace amounts of fine-grained magnetite (Hidalgo et al., 2000). The gangue is composed of pyrite, quartz, carbonate and mica.

The deposit is accompanied by pervasive sericitic and chloritic hydrothermal alteration of the host rock around the body, with a quartz alteration zone in the central region of the stockwork (Bobrowicz, 1995).

**2.3.2 Host stratigraphic sequence**

The stratigraphic sequence hosting the Aguas Teñidas deposit belongs entirely to the VSC. It is dominated by volcanic and subvolcanic rocks, with minor sedimentary (shales) materials (Bobrowicz, 1995; McKee, 2003; Conde, 2016; Conde and Tornos, 2019) (Fig. 5). The volcanostratigraphic sequence in the Lomero Poyatos and Aguas Teñidas areas was subdivided into 6 tectonostratigraphic units separated by tectonic contacts by Conde (2016) and Conde and Tornos (2019); their

characteristics are summarized in Table 1. The Footwall Felsic Unit and Upper Felsic Unit were interpreted to belong to a single felsic volcanic complex which was dismembered by tectonics based on facies and geochemical characteristics. In addition, it was suggested that the sequence was tectonically inverted, with the Andesite Unit representing the oldest one (Conde and Tornos, 2019).

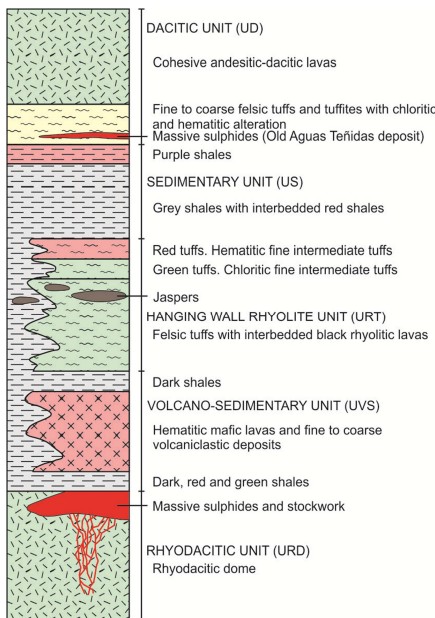


**Figure 5: Local stratigraphy at Aguas Teñidas deposit, not to scale. Nomenclature follows original description by MATSA. Modified from MATSA.**

Aguas Teñidas deposit is interpreted to have formed by replacement of the permeable and reactive uppermost autobrecciated

and partially devitrified facies of a dacitic dome (Bobrowicz, 1995; Tornos, 2006) of the Footwall Felsic Unit (Conde, 2016; Conde and Tornos, 2019). The main lithologies of the hanging wall are red lavas (predominant in the northern part of the deposit), red to purple volcaniclastic rocks, and red/green metapelites (present in the southern part of the deposit) which are characterized by strong vertical and lateral facies changes (Hidalgo et al., 2000) (Fig. 5). The host felsic dome was named



URD (Unidad Riodacítica; Rhyodacite Unit) by the mining company (MATSA); this local name will be used in this work to
differentiate it from the broader Footwall Felsic Unit and Upper Felsic Unit. Similarly, local names of the Hanging wall
Felsic Unit will be also used. The equivalences between unit names used in previous works from Aguas Teñidas (as shown
in Fig. 5) and those in Conde and Tornos (2019) is provided in Table 1.

| Unit | Lithology | Thickness |
|---|---|---|
| Andesite U. (Dacitic Unit, UD) | Andesitic dome complexes rich in hyaloclastite breccias and andesitic volcaniclastic rocks. Less felsic dykes and breccias, and intercalated shales | 100-200 m |
| Upper Felsic U. (no equivalence) | Dacitic to rhyolitic dome complexes with characteristics equivalent to Footwall Felsic Unit | 100-150 m |
| Sedimentary U (Sedimentary Unit, US). | Grey siltstone with interlayered shale and fine-grained epiclastic rocks | < 150 m |
| Hanging wall Felsic U. (Hanging wall Rhyolite Unit, URT) | Coherent rhyolitic domes and associated volcaniclastic rocks intercalated with polymictic sedimentary rocks, cut by mafic to felsic sub-volcanic intrusions | 250-300 m |
| Volcano-sedimentary U. (Volcano-Sedimentary Unit, UVS) | Vesicular basaltic lava and associated epiclastic sandstone and siltstone, intercalated with thin shale beds | < 300 m |
| Footwall Felsic U. (Rhyodacite Unit, URD) | Feldspar-quartz-phyric rhyodacite (crypto-)dome complexes (massive and associated coarse proximal brecciated and finer-grained distal volcaniclastic facies), with sills and dykes of similar composition | 200-400 m |

**Table 1. Teconostratigraphic units in the Lomero Poyatos – Aguas Teñidas zone as defined by Conde and Tornos (2019). Local**
**names used by MATSA and previous works in Aguas Teñidas are given in brackets, with the original acronyms in Spanish for**
**reference.**

As in other deposits in the northern IPB, the rocks hosting the Aguas Teñidas deposit underwent three stages of
alteration/modification: 1) metasomatism/alteration of volcanic rocks by interaction with seawater during and soon after
emplacement in submarine conditions, which transformed basalts into spilites and felsic rocks into keratophyres and quartz-
keratophyres (Munhá and Kerrich, 1980); 2) hydrothermal alteration related to the mineralizing event; 3) deformation and
metamorphism (prehnite-pumpellyite facies) (Bobrowicz, 1995; Sánchez-España et al., 2000; McKee, 2003) and related
remobilization.

### 3. Methods

The investigation of geochemical and mineralogical vectoring tools at Aguas Teñidas deposit has been performed through
the study of samples collected from representative drill cores provided by MATSA mining company.

### 3.1 Sampling

Sampling was aimed at collecting samples from proximal, medial, and distal host rocks to the massive sulphides, as well as
from shallow, medial, and deep regions of the stockwork in order to characterize the lithological background as well as
variations with proximity to ore and within the hydrothermal system. 551 samples were collected from 12 drill cores. A list
of the studied drill cores and the purpose behind their sampling is provided in Table 2. Their location and relationship to
mineralization is shown in Figures 3 and 6.

### 3.2 Analytical methods

171 representative samples were selected for whole rock geochemical analysis. Sample preparation and analysis of major
and trace elements were performed commercially by SGS (Société Générale de Surveillance). Samples were powdered to



85% passing 75 µm mesh in a Cr-free steel mill. Major elements were analysed by X-Ray Fluorescence on glass disks prepared by borate fusion. Trace elements were analysed by Inductively Coupled Plasma Atomic Emission Spectroscopy (ICP-AES) and Inductively Coupled Plasma Mass Spectrometry (ICP-MS) on samples prepared by Na$_2$O$_2$/NaOH fusion followed by dissolution in nitric acid.

| Drill core | Location | Characteristics | Purpose of sampling |
|---|---|---|---|
| MU-2 | Distal , > 900 m horizontal distance from the massive sulphides. Drilled from surface | In its lowermost portion it intersects lavic, pyroclastic and epiclastic deposits other than the host unit (URD) at its approximately equivalent stratigraphic/structural position | Sampled in its lowermost portion to study the petrological and geochemical characteristics of lithologies in distal locations to the ore in order to establish the non-mineralized background characteristics |
| MU-5 | Distal, close to MU-2. Drilled from surface | Stratigraphy equivalent to MU-2, its lowest portion intersects the top of URD | Samples were taken from URD and immediately overlying volcanic rocks to establish background characteristics |
| AM-49 | Marginal, westernmost end of the massive sulphides of Aguas Teñidas deposit. Drilled from surface | It intersects the old Aguas Teñidas orebody and only 5 cm of the massive sulphides of the currently exploited orebody, which presents no underlying stockwork at this location, reaching URD unit | Sampling aimed at the study of the host unit (URD) and overlying deposits immediately around the massive sulphides level in marginal positions. In addition, samples were taken from shallower regions of the drilling to characterize other units in the stratigraphic sequence |
| AGI-888 | Drilled from an underground gallery in the central area of the western body of Aguas Teñidas. | It proceeds stratigraphically upwards, from regular host unit (URD), through the stockwork, across the massive sulphides, into the structurally/stratigraphically overlying deposits | Samples were taken from all sections to study mineralogical and geochemical variability around an area with thick massive sulphides and stockwork development. |
| AGI-808 | Equivalent to AGI-888 | Equivalent to AGI-888 | Equivalent to AGI-888 |
| DST-332 | Drilled from an underground gallery near the eastern part of the western body of Aguas Teñidas | It starts in the regular URD unit and proceeds through the stockwork in a downwards direction from shallower to deeper portions of the stockwork system | Sampling aimed at characterizing the stockwork system and the chemistry of host rocks in its central parts |
| ATS-7 | Drilled from an underground gallery close to the central part of the eastern body of Aguas Teñidas | It runs vertical through the URD nearly parallel to the stockwork | Sampling aimed at studying the footwall host rock at a close distance to the stockwork |
| DST-386 | Drilled from an underground gallery immediately east of the eastern end of the massive sulphides | It starts N of the Northern fault and crosses it into the stockwork, advancing downwards to the deep areas of the stockwork system | Sampling aimed at the study of the stockwork in an area complementary to that studied in the western sector of the deposit |
| DST-392 | Equivalent to DST-386 | Equivalent to DST-386 but drilled upwards | Sampling aimed at the study of the stockwork |
| AE-68 | Proximal drill core at a close distance (ca. 250 m) NE of the eastern end of the massive sulphides. Drilled from surface | It transects 340 m of the overlying sequence before reaching the top of the host unit (URD), which is drilled for 200 m. It intersects no massive sulphides, but near its lower end it crosses a weak stockwork with disseminated sulphides and minor veins | Sampling focused on: 1) the host unit to characterize the distal stockwork system and the petrological and geochemical variability of the host unit at different distances from it; 2) the volcaniclastic units immediately overlying the host unit to explore possible influences of the mineralizing process; 3) other lavic, pyroclastic and epiclastic units in shallower regions of the drilling to characterize the lithological variability in the stratigraphic sequence |
| AE-69 | Equivalent to AE-68 | Equivalent to AE-68 | Equivalent to AE-68 |
| AE-70 | Equivalent to AE-68 | Equivalent to AE-68 but does not reach the distal stockwork | Equivalent to AE-68 |

**Table 2. Drill cores studied and sampled in this work**



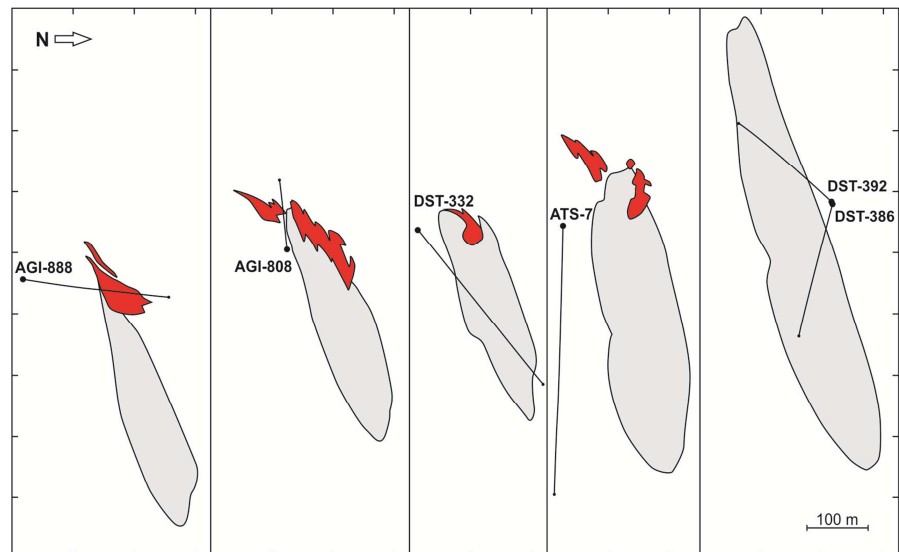

**Figure 6: Trajectories of studied drill cores in cross-sections perpendicular to the massive sulphides orebody and feeder zone elongation. Legend follows Figure 3. Data provided by MATSA.**


Portable XRF analyses were performed on a selection of samples, on hand specimen as well as on pressed pellets. A Thermo NITON XL3t 900Analyzer with GOLDD Technology was used at the facilities of the Geosciences Institute of the Spanish Research Council and Complutense University of Madrid. Prior to sample analysis, an assessment of the performance of our device was made, particularly on the effect of equipment warm-up, measuring time, distance to sample, water content, and

number of analysis per sample following the examples and recommendations of previous works (e.g. Ge et al., 2005; Hall et al., 2013, 2014; Bourke and Ross, 2015; McNulty et al., 2020; Laperche and Lemière, 2021); results and discussion of these aspects are presented in Supplementary Material 1.3. Measurements were made using the Cu/Zn mining mode with 30 s analysis time per filter (4 filters) after an initial 15 minute warm-up time at the beginning of each session. Equipment calibration was performed by calculating calibration lines from the measurement of pressed powder pellets (15 g of sample

pressed at 200 kN for 2 minutes with no binding materials such as resin or wax) of samples previously analysed for whole rock geochemistry at SGS. These pellets were prepared with the same powder used for whole rock geochemical analysis. 15 samples representative of all lithologies and compositions of Aguas Teñidas, plus 2 additional shale samples from the southern IPB were used. Pellets used for calibration were regularly measured during equipment operation to check for measurement consistency and equipment drift.


Thin sections of 117 representative samples were prepared for petrographic study and mineral chemistry analysis.

## 4. Results and discussion

### 4.1 Mineralogical zoning

The mineralogical zoning in alteration halos around VMS deposits has long been used as an empirical vectoring tool

worldwide (e.g. Large et al., 2001a and references therein for Australian deposits). At Aguas Teñidas it was first studied and described from the observation of the eastern sector of Aguas Teñidas by Bobrowicz (1995), Hidalgo et al. (2000), McKee et al. (2001), and McKee (2003), who focused on the footwall. The study of new drill cores in this work confirms their



observations and allows extending them to the western sector of the deposit. In addition, we identify albitic alteration in the hanging wall for the first time.


The samples from distal cores (MU-2, MU-5) are beyond the influence of hydrothermal alteration related to Aguas Teñidas deposit, thus providing information on the seafloor metasomatism that dominates the geological background around it. In volcanic and subvolcanic materials the alteration assemblage is largely controlled by the original rock composition (mafic v. felsic), whereas the degree of alteration is larger in volcaniclastic rocks compared to cohesive lavas. Both mafic (Fig. 7a) and

felsic (Fig. 7b) rocks show complete feldspar albitization in the less altered rocks, which progresses to incipient sericitization and formation of chlorite patches in the most altered samples. In mafic rocks the alteration assemblage is completed by chlorite and epidote, whereas in felsic ones it is dominated by muscovite and quartz. Patches of fine-grained chlorite in the groundmass are interpreted as indicative of former mafic crystals, as no mafic minerals are preserved in the studied rocks. Original igneous quartz phenocrysts in felsic rocks (e.g. in URD unit) are typically preserved, although variably modified by

dissolution and/or overgrowth of epitaxial quartz.

In a more proximal setting (e.g. AE-68, AE-69, AE-70) sericite-quartz alteration dominates within the URD footwall, with alteration degree increasing from background lithologies towards the centre of the system. In less altered rocks feldspar phenocrysts pseudomorphs evidenced by coarser muscovite crystals may be preserved within a finer-grained alteration

groundmass (Fig. 7c). However, further alteration completely obliterates the original rock texture except for modified remnants of quartz phenocrysts. This alteration assemblage also dominates the distal stockwork (e.g. AE-68, AE-69), where additional carbonate alteration may also occur (Fig. 7d), and external parts of the proximal stockwork (e.g. AGI-888, AGI-808, DST-332, DST-386). In the central parts of the stockwork system observed in DST-332 and DST-386, sericite-quartz alteration transitions to chlorite-quartz alteration (Fig. 7e, f), which is most intense in areas presenting chalcopyrite in the

sulphide assemblage. In the studied drill cores from this part of the system there are no lithologies other than the URD, and therefore the effect of proximal alteration on them is unknown.

The reconstruction of the geometry of the alteration zones in the footwall of Aguas Teñidas shows a pervasive, asymmetric, elongated and concentric cone-shaped hydrothermal alteration which bears the stockwork and which passes laterally, from

core to edge, from quartz (not observed in this study), to chlorite, sericite–chlorite, and sericite zones (Bobrowicz, 1995; Hidalgo et al., 2000; McKee et al., 2001), all of them with quartz as an alteration phase (Fig. 4). In the upper parts of the chlorite zone, particularly along its northern and southern contacts with the siliceous zone, chlorite-carbonate alteration zones are also found (Bobrowicz, 1995). Siliceous alteration zones at the centre of the system are not continuous along the deposit; this is considered to indicate non-uniform supply of hydrothermal fluids along the feeder system, with location(s) of

higher intensity (McKee, 2003). Hydrothermal alteration transitions to seafloor metasomatism characteristics at the margins of the system.

Even though the hanging wall to the deposit is mostly tectonically emplaced over the ore, a proximal sericite alteration zone followed by an albite one in more distal positions have been observed in volcaniclastic rocks of core AGI-888 (Fig. 11). The

sericite alteration zone consists of a fine-grained assemblage of muscovite, quartz and minor to rare chlorite, and occurs in rocks of felsic, intermediate and mafic compositions (samples 179.8, 180.9, 191.5 and 206.9). This indicates a stronger control of the hydrothermal alteration and weaker control of the original rock composition compared to zones with dominant seafloor alteration. Albite alteration was observed in a sample from depth 238.7, within volcaniclastic rocks of mafic composition (Fig. 7g). Contrary to albite in distal lithologies, which forms by albitization of igneous feldspars, albite in this

alteration zone occurs as a fine-grained groundmass also containing muscovite and minor amounts of chlorite (Fig. 7h).

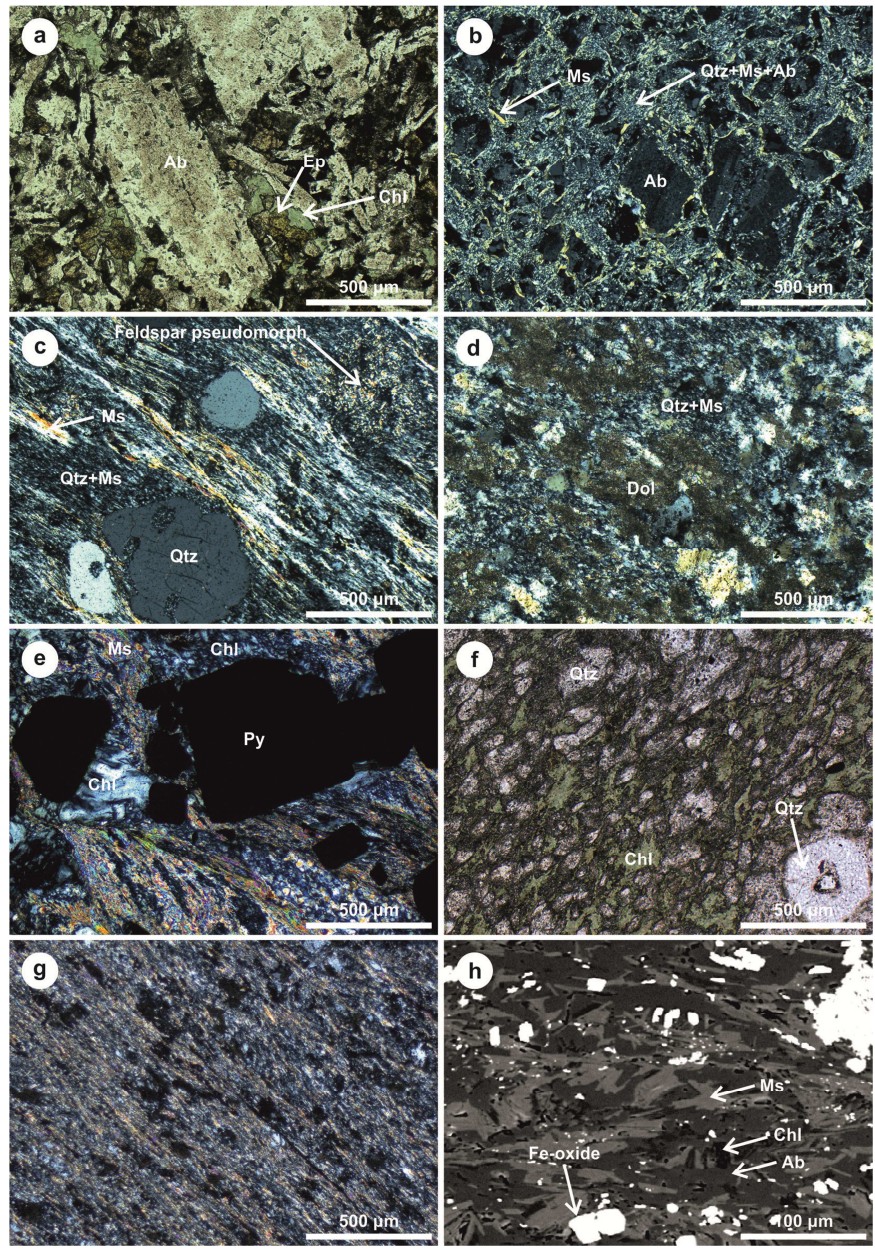

**Figure 7:** Petrographic microscope pictures of altered rocks in the host sequence of Aguas Teñidas deposit. (a) Distal volcanic rock of intermediate composition (MU-2/895.5) with seafloor alteration; plane polarized light. (b) Distal URD (MU-5/875.25) with seafloor alteration; cross polarized light. (c) Hydrothermally altered URD in the sericite alteration zone (AE-68/367.6); cross polarized light. (d) URD in the distal stockwork zone (AE-69/384.0), with quartz-sericitic and carbonatic alteration; cross polarized light. (e) URD in the medial part of the central stockwork (DST-332/139.7), with quartz+sericite+chlorite alteration; cross polarized light. (f) URD in the central part of the central stockwork (DST-332/251.5), with chlorite+quartz alteration; note the preserved quartz phenocrysts; plane polarized light. (g) Fine mafic volcaniclastic rock in the hanging wall oxidized albitic alteration zone (AGI-888/238.7); it consists of a fine-grained muscovite+albite+minor chlorite+iron oxides assemblage; cross polarized light. (h) Detail of the fine-grained mineral assemblage in sample AGI-888/238.7; backscattered electrons scanning electron microscope image. Ab: albite; Chl: chlorite; Dol: dolomite; Ep: epidote; Ms: muscovite; Py: pyrite; Qtz: quartz.





These observations indicate that thrusts and shear zones at the top of the massive sulphides at Aguas Teñidas were affected by minor displacements, which were insufficient to decouple the orebody and its associated hydrothermally originated hanging wall alteration.

In addition to sericitic and albitic alteration, the hanging wall to Aguas Teñidas shows a pervasive overprinted oxidizing alteration (Hidalgo et al., 2000; Tornos, 2006) which controls the colour of the hanging wall unit (Fig. 7h). Iron oxides (magnetite and hematite) replaced pyrite in this unit in zones adjacent to or inside the shear band (Tornos, 2006). Tornos (2006) describes that rocks within this structure show evidence of syn-deformational oxidation (the shale is purple and fragments and lenses of reddish silicified rocks are common), which is interpreted as suggesting that the oxidized fluids percolated along these structures during the Variscan orogeny and, thus, that at least part of this hanging wall oxidation was tectonically related.

Footwall mineralogical zoning at Aguas Teñidas is equivalent to that found in most volcanic-hosted deposits in the northern Iberian Pyrite Belt, which present an innermost quartz-rich zone (not present in all deposits), followed by chlorite-rich and sericite-rich zones (Relvas, 1990; Costa, 1996; Tornos, 2006). An additional ultrapheripheral alteration zone with Na-rich mica + quartz ± disseminated pyrite was observed at Gaviao orebody by Relvas et al. (1990) and Barriga and Relvas (1993) up to 1000 m away from the orebody. In contrast, hydrothermal alteration in the predominantly shale-hosted deposits of the southern IPB is usually dominated by the chloritic type due to the lithological control (Tornos et al., 1998; Ruiz et al., 2002). Carbonate alteration is common throughout the IPB (e.g. at Rio Tinto, Tharsis, La Zarza) and typically occurs in marginal zones of the massive sulphides, at the interphase between the sulphides and the underlying stockwork, as independent veins in the stockwork and as disseminations (Williams et al., 1975; Strauss et al., 1981; Tornos et al., 1998). Hanging wall alteration has been poorly characterized in the IPB due to the commonly thrusted character of the hanging walls currently located on top of the massive sulphides bodies (Tornos, 2006; Martin-Izard et al., 2016); Aguas Teñidas deposit provides a good example for the understanding of this part of the system.

Mineralogical zoning in the IPB follows that of most typical VMS systems worldwide (Large et al., 2001a; Franklin et al., 2005; Gibson et al., 2007, Soltani-Dehnavi et al., 2018a) (Fig. 1). Hanging wall alteration halos, which are mostly lost in the IPB, are usually minor and dominated by sericite-rich alteration, with local zones of albitic alteration (e.g. Large et al., 2001a), as observed at Aguas Teñidas. Remarkably, in the Bathurst Mining Camp an outermost albite-Mg-chlorite alteration zone has been described both in the hanging wall and footwall (Soltani-Dehnavi et al., 2018a and references therein). The extent of alteration haloes tends to correlate to the size of the deposit, with sericite-rich alteration halo typically extending up to hundreds of meters away from the massive sulphides (e.g. Large et al., 2001a). In the Gaviao orebody of the Aljustrel deposit in the IPB, sericite-rich alteration has been described up to 500 m away from mineralization (Relvas et al., 1990), which is a distance similar to that observed at Rosebery, a replacive-type deposit in the Mount Read Volcanics Belt, Tasmania (Large et al., 2001b).

Once the mineralogical zoning around a specific hydrothermal system is understood, fast portable mineralogical characterization techniques such as hyperspectral equipment can be used to detect mineralogical changes (e.g. Herrmann et al., 2001; Ross et al., 2019; Hollis et al., 2021) in nearly real time at minimal expense during ongoing exploration, which allow for targeting decisions to be made rapidly.



### 4.2 Whole rock geochemistry

The new whole rock geochemistry data obtained in this study are provided in Supplementary Material 2. These have been
used to investigate: 1) the composition of the host unit; 2) the behaviour of major elements during hydrothermal alteration;

3) the trace element geochemical haloes around the deposit.

### 4.2.1 Characterization of the host unit

Identification of mineralization-related lithologies (e.g. specific lithological units) - or lithologies indicative of favourable
environmental conditions - based on tools such as whole rock geochemistry is traditionally used as a pathfinder in well
characterized host stratigraphic sequences in VMS (e.g. Barret et al., 2005; Schlatter, 2007) and other mineral systems (e.g.
SEDEX, Rieger et al., 2021).

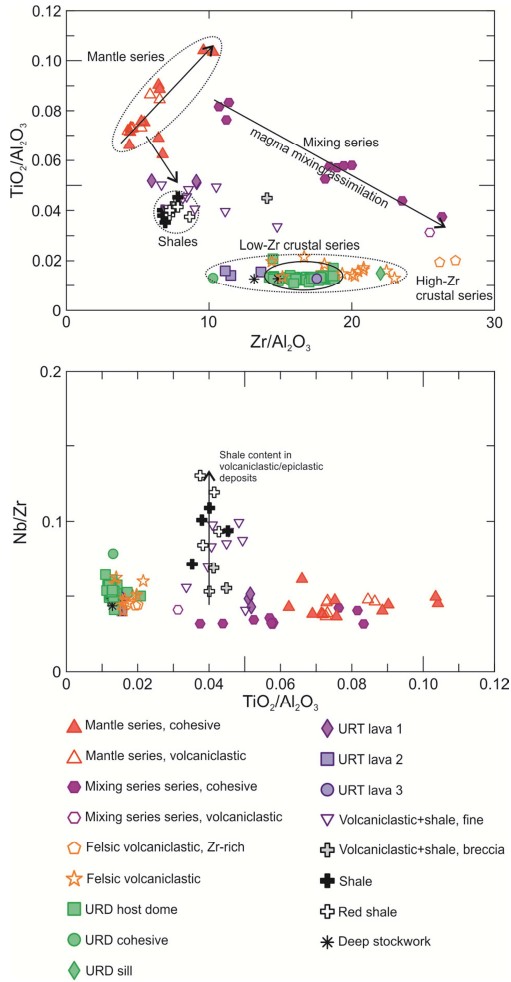

**Figure 8: Whole rock geochemistry discrimination diagrams of rocks in the Aguas Teñidas host stratigraphic sequence based on**
**immobile elements. Ratios calculated from major (oxides in wt %) and trace (µgg$^{-1}$) element contents.**

The chemical characteristics of lithologies in the stratigraphic sequence hosting Aguas Teñidas deposit have been studied
using whole rock geochemistry of immobile major and trace elements. Discrimination diagrams have been elaborated based



on ratios of Al, Ti, Zr and Nb, which typically present an immobile character under hydrothermal regimes similar to those
forming VMS systems (Floyd and Winchester, 1978; MacLean and Kranidiotis, 1987). These diagrams allow recognizing
specific lithologies, including the host unit to the massive sulphides (Fig. 8). This approach has been shown to be effective
for identification and correlation purposes in other VMS districts (e.g. Barret et al., 2005; Schlatter, 2007) as well as in the
study of altered volcanic-dominated stratigraphic sequences in other settings (e.g. Winchester and Floyd, 1977; Gisbert et al.,
2017).


The lithologies studied at Aguas Teñidas include cohesive lavic rocks (as lava domes, lava flows, and sills), breccias of
volcanic clasts hosted in shale, coarse volcaniclastic deposits, fine volcaniclastic/epiclastic deposits with variable content in
shale, and shales. Samples in Figure 8 have been grouped according to their lithology, whole rock geochemistry and unit -
the latter only in the case of cohesive volcanic rocks in Footwall Felsic Unit and Hanging wall Felsic Unit, which correspond
to URD and URT units in the mine stratigraphy. Volcanic rocks are subalkaline in composition, as indicated by low Nb/Y
(Fig. 9, Pearce, 1996) and Nb/Zr (Fig. 8b) ratios. This is consistent with the composition of volcanic rocks in the IPB, where
only minor alkaline rocks have been found (Munhá, 1983; Mitjavila et al., 1997; Thieblemont et al., 1997). The two main
compositional clusters in Fig. 9 correspond to the mantle-derived basaltic tholeiitic magmas and crust-derived felsic magmas
described by Mitjavila et al. (1997).


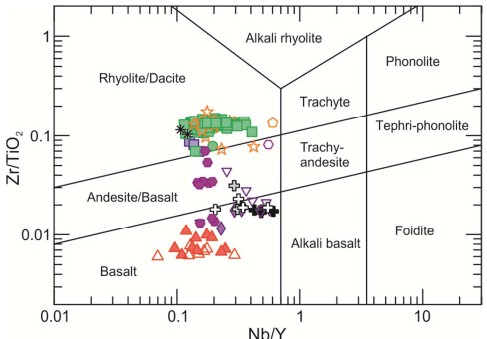

**Figure 9: Zr/TiO$_2$ v. Nb/Y diagram (Pearce, 1996, after Winchester and Floyd, 1977). Ratios calculated using concentrations in µgg$^{-1}$. Symbols as in Figure 7.**

The TiO$_2$/Al$_2$O$_3$ v. Zr/Al$_2$O$_3$ discrimination diagram (Fig. 8a) is the most useful one at Aguas Teñidas. Three main
compositional groups are recognized in it: 1) high-TiO$_2$/Al$_2$O$_3$ low-Zr/Al$_2$O$_3$ mantle-derived volcanic rocks; 2) low-
TiO$_2$/Al$_2$O$_3$ crust-derived volcanic rocks; and 3) shales. In addition, mixed compositions between them exist. High-
TiO$_2$/Al$_2$O$_3$ low- Zr/Al$_2$O$_3$ volcanic rocks depict a linear trend which is interpreted as a fractional-crystallization-controlled
mantle-derived tholeiitic magma differentiation trend based on the positive correlation between TiO$_2$/Al$_2$O$_3$ and Zr/Al$_2$O$_3$
(Fig.8a), constant Nb/Zr (sensitive to mantle partial melting degree), and deviation from mixing trends with crustal material
(e.g. shales or compositions equivalent to felsic magmas) which could indicate assimilation or magma mixing. Low-
TiO$_2$/Al$_2$O$_3$ volcanic rocks present a wide range of Zr/Al$_2$O$_3$, from 10 to over 25 in the analysed samples, but reaching over
50 in other rocks from the area (e.g. Conde, 2016; Conde and Tornos, 2019). This large range may be due to differences in
source rock composition (suggested also by differences in Nd isotope compositions in other areas, e.g. Valenzuela et al.,
2011, Donaire et al., 2020), variable degrees of partial melting of refractory phases during magma formation (e.g. Rosa et al.,
2004, 2006: de Oliveira et al., 2011), and/or to magma evolution (e.g. zircon fractionation) prior to emplacement (e.g. Barret
et al., 2008). In addition to these two groups, intermediate cohesive lava compositions mark magma mixing trends of mantle-
derived magmas with crustal ones, both from the high-Zr/Al$_2$O$_3$ end (e.g. "mixing series" in Fig. 8a) and low-Zr/Al$_2$O$_3$ one





(e.g. URT lava 1). Mixing of fine-grained volcaniclastic/epiclastic volcanic products with shale is common; these rocks

show intermediate $Ti/Al_2O_3$ contents (Fig. 8a) and higher $Nb/Zr$ ratios (Fig. 8b).

The Footwall Felsic Unit, which hosts the mineralization, and Hanging wall Felsic Unit are both dominated by crust-derived low-$Zr/Al_2O_3$ cohesive lavas with minor interbedded/intruded lavas from the mantle and mixing series (Fig. 8a). Within the Footwall Felsic Unit, the lava dome that hosts the massive sulphides ("URD host dome" in Fig. 8a) shows a restricted

compositional range despite its large dimensions (> 2 km long). Below it, drillings have intersected the top of another dome; only one sample ("URD cohesive"), which presents lower $Zr/Al_2O_3$ compared to the host dome, has been yet analysed. A felsic sill currently within the massive sulphides ("URD sill") intruded the host unit likely prior to massive sulphide formation according to facies relationships. This sill presents higher $Zr/Al_2O_3$, and shows only minor replacement by massive sulphides, which may be due to different texture and/or composition relative to the host lava. From the Hanging

wall Felsic Unit, which was interpreted to be equivalent to the Footwall Felsic Unit by Conde (2016) and Conde and Tornos (2019), three cohesive felsic lavas were analysed for comparison. "URT lava 2" and "URT lava 3" are indeed similar in composition to rocks from the Footwall Felsic Unit, within the field of low-$Zr/Al_2O_3$ rocks. Whereas "URT lava 3" falls within the composition of the host dome, "URT lava 2" falls outside the compositional field of the host dome, between its lower $Zr/Al_2O_3$ end and the composition of the dome below the host dome ("URD cohesive"). On the other hand, "URT lava

1" shows a significantly different composition, as it is within a possible mixing line between the lower ends of mantle- and crustal-derived magmas trends.

These discrimination diagrams can also be used in the study of altered samples and track the hosting unit even in highly altered zones. Two samples from the chloritic alteration zone at the core of the deep stockwork of Aguas Teñidas were

analysed ("Deep stockwork" in Fig. 8). The original petrographic characteristics of these rocks have been completely erased. However, their whole rock geochemistry indicates that they belong to the Footwall Felsic Unit, likely the host dome (URD, sample DST-332/251.5) or the underlying one (DST-332/275.9) (Fig. 8a). This is consistent with the presence of the host dome around the heavily altered stockwork, and confirms the usefulness of the chosen immobile trace elements. Thus, discrimination diagrams presented in this work can be used with confidence for the identification of lithological units within

the stratigraphic sequence of Aguas Teñidas deposit.

Works dealing with a detailed lithogeochemical characterization of volcanic units hosting and surrounding specific orebodies such as the one presented here for Aguas Teñidas are scarce in the IPB (e.g. Barret et al., 2008; de Oliveira et al., 2011) as broader studies have been typically performed (e.g. Mitjavila et al., 1997; Sánchez-España et al., 2000; Rosa et al., 2004,

2006; Valenzuela et al., 2011, Conde and Tornos, 2019). However, similarly to what has been observed at Aguas Teñidas, available detailed studies usually show the presence of several felsic volcanic units which can be identified based on immobile element ratios like those used here (e.g. $Zr/Al_2O_3$). For example, at Feitais orebody (Aljustrel), Rhyolites A, B, C and X show different compositional ranges ($Zr/Al_2O_3$ ca.14-16, 17-20, 10-13, and 25-26, respectively) (Barret et al., 2008). Similarly, at Lagoa Salgada the host feldspar- and quartz-phyric rhyodacite hosting the massive sulphides has a $Zr/Al_2O_3$

ratio of ca. 5.5 to 8, whereas for the barren quartz-phyric rhyodacite this ratio is ca. 8 to 11 (de Oliveira et al., 2011). At the deposit scale this approach can thus be highly useful during deposit exploration and characterization in the heavily tectonized IPB as a vector to locate the mineralized stratigraphic horizon within a previously geochemically characterized sequence. Available detailed studies are insufficient, though, to analyse patterns which could be extrapolated for a wider use within the IPB in terms of inferring the barren or fertile character of a given unit according to its whole rock geochemistry.

As a first approach, though, it seems that, as seen at Aguas Teñidas as well as at Feitais and Lagoa Salgada examples, VMS deposits in the IPB are typically related to low-Zr felsic magmas (Rosa et al., 2004, 2006; Valenzuela et al., 2011; Donaire et



al., 2020; Conde and Tornos, 2019). In addition, data from Rio Tinto-Nerva and Paymogo Volcano-Sedimentary Alignment areas indicate that these magmas also present less radiogenic Nd isotope signatures, which has been interpreted as resulting from shallower partial melting of more evolved crustal rocks (Valenzuela et al., 2011; Donaire et al., 2020). Additional work

is needed to confirm these observations.

### 4.2.2 Vectors based on major elements

Hydrothermal alteration occurs in open system conditions, where changes in rock geochemistry occur due to supply or removal of mobile elements (Franklin et al., 2005; Hanington, 2014). Chemical changes can be tracked by observing variations within the system in individual elements contents (e.g. gains and losses, usually calculated based on methods such

as Pearce element ratios or the isocon method, Pearce, 1968; Grant, 1986; e.g. Madeisky and Stanley, 1993; Barret et al., 2005; Dong et al., 2017), ratios between elements (e.g. Na/S, Large et al., 2001a), or commonly used indicator indexes such as the Alteration Index (AI, Ishikawa et al., 1976) and the Chlorite-Carbonate-Pyrite Index (CCPI, Large et al., 2001c) (e.g. Piercey et al., 2008; Dong et al., 2017). These variations can be used as vectors to ore (e.g. Madeisky and Stanley, 1993; Large et al., 2001b).


Volcanic rocks around Aguas Teñidas deposit have undergone two main stages of alteration: seafloor alteration and mineralization-related hydrothermal alteration (Bobrowicz, 1995; Sánchez-España et al., 2000; McKee, 2003). Samples from distal portions of the system (drill cores MU-2 and MU-5) show the effects of seafloor alteration. In mafic rocks from the mantle series, feldspars (presumably plagioclase in origin) are typically completely albitized, and partially replaced by

chlorite or sericite. Mafic minerals are completely replaced by chlorite and epidote, and the groundmass consists of an assemblage dominated by fine-grained chlorite, epidote and minor carbonate. Whole rock geochemical changes associated to this alteration produce little shifts in the position of these rocks within the alteration box plot (Large et al., 2001c) (Fig. 10). In felsic rocks from crustal origin seafloor alteration is dominated by albitization and variable sericitization of feldspars and groundmass, and chloritization of the much scarcer mafic phases. The overall chemical effect is a variable shift towards the

albite, sericite and chlorite poles of the alteration box plot (Fig. 10).

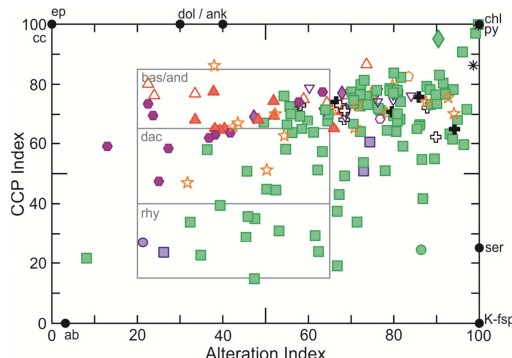

**Figure 10: Alteration Box Plot (Large et al., 2001c); Chlorite-Carbonate-Pyrite Index (CCPI) is defined as 100(FeO + MgO)/(FeO + MgO + K₂O+ Na₂O), Alteration Index (AI) of Ishikawa et al. (1976) is defined as 100(MgO + K₂O)/(MgO + K₂O + CaO + Na₂O).**
**The fields of rhyolitic (rhy), dacitic (dac) and basaltic and andesitic (bas/and) least altered volcanic rocks as described by Large et al. (2001) based on data from Rosebery, Que River, and Hellyer areas of the Mount Read Volcanics are depicted. ab: albite; ank: ankerite; chl: chlorite; dol: dolomite; ep: epidote; K-fsp: K-feldspar: py: pyrite; ser: sericite. Symbols as in Figure 7.**

Hydrothermal alteration adds to the previous modifications induced by seafloor alteration. The main effect is an increase in
both Alteration and CCP Indexes caused by further sericitic and chloritic alteration, and by pyrite precipitation, which increase towards the centre of the system. Additional local variable carbonate alteration may occur. Note that changes in



silica content - e.g. resulting from silicification, which is pervasive in most parts of the hydrothermal system in Aguas Teñidas - are not considered in the plot. These modifications are shown in the alteration box plot as a compositional convergence towards the chlorite and pyrite vertex (AI = 100, CCPI = 100) regardless of the original rock composition.

Samples closest to that vertex are those from the nearly completely chloritized rocks (with or without accompanying silicification) collected from the chlorite alteration zone or from locally chlorite- or pyrite-rich bands and veins. These correspond to samples from the sill within the massive sulphides (AGI-888/162.5), or the central parts of the stockwork in distal (AE-68), shallow (AGI-888), or deep (DST-332) zones. Higher carbonate contents in samples from areas with carbonatic alteration (e.g. distal stockwork in AE-69, URD host dome) result in higher CCPI at the same AI.


Downhole CCPI and AI variations are represented in Figure 11 (AGI-888 and AE-69 as examples) and Supplementary Materials 1.4. In AGI-888, which crosses the massive sulphides, the maximum AI and CCPI values occur by the massive sulphides and decrease away from it, representing useful vectors to ore. However, CCPI values in the hanging wall mafic hematitic tuffs are higher than in the immediately underlying felsic tuffs and tuffites and footwall URD, despite their more

distal location relative to mineralization. This is due to the original mafic composition of host rocks (higher initial FeO and MgO contents), which highlights the importance of considering the original rock composition when working with chemical indexes. The samples analysed in AE-69 belong mostly to the URD host dome. In the higher section of URD, index values are constant, becoming more variable around the distal stockwork further down. In this lower area, within this larger variability, AI values slightly increase, while CCPI decreases slightly; this is due to a decrease in MgO and Na$_2$O coupled

with an increase in K$_2$O which will be discussed later. A higher variability of the alteration indexes around stockwork areas compared to more regular behaviour in distal portions is also seen in ATS-7 and AE-68. This may reflect the more pervasive and homogeneous character of seafloor alteration compared to more focused and permeability-controlled (e.g. related to stockwork structures) hydrothermal alteration.

Element mass changes were investigated using the isocon method (Grant, 1986), which requires studied rocks to have had a common original chemical composition before alteration (Grant, 1986; Grant, 2005). Thus, analysis was restricted to the URD unit as it is the only compositionally homogeneous lithological unit for which least altered to heavily altered compositions were available. After data examination, TiO$_2$, Al$_2$O$_3$, Nb, Ta, Th, and Zr were chosen as reference immobile elements for isocon calculation, and no normalization factors were applied. Sample MU-5/875.25, which is representative of

the URD host dome in the most distal sampled area, was used as the least altered composition. Complete results of this analysis are provided in Supplementary Material 2. $\Delta C_i$ of MgO, Na$_2$O and K$_2$O are depicted as examples in downhole diagrams in Figure 11 and Supplementary Material 1.4. $\Delta C_i$ represents the absolute difference between the actual concentration of a given element in a rock, and the concentration it would have had if it had behaved as immobile. It therefore indicates element mass gains and losses in concentration units (wt. % for major elements, ppm for trace elements).


Establishing general trends with distance to ore is difficult given the sampling approach followed in this study. However, the comparison of distal samples with those of the sericite to chlorite alteration zones shows that there is a remarkable generalized mass gain of SiO$_2$ in the area around the Aguas Teñidas deposit, which is accompanied by a smaller gain of MgO and FeO relative to the distal samples, and variable loss or local minor enrichment of Na$_2$O and K$_2$O. In marginal

positions (upper URD zones in cores AE-68 and AE-69), FeO, MgO, Na$_2$O and K$_2$O show a rather constant behaviour in profiles across the host dome above the stockwork zone, indicating a fairly constant effect of seafloor metasomatism throughout the lava dome in this area. K$_2$O depletion and slight Na$_2$O and MgO enrichment in these areas relative to the reference sample likely indicate slightly higher seawater controlled albitization and sericitization±chloritization, which may



be related to local variability of regular seafloor metasomatism and/or represent enhanced metasomatism under the influence

of the hydrothermal system (this area occurs within the weak sericitic alteration zone).

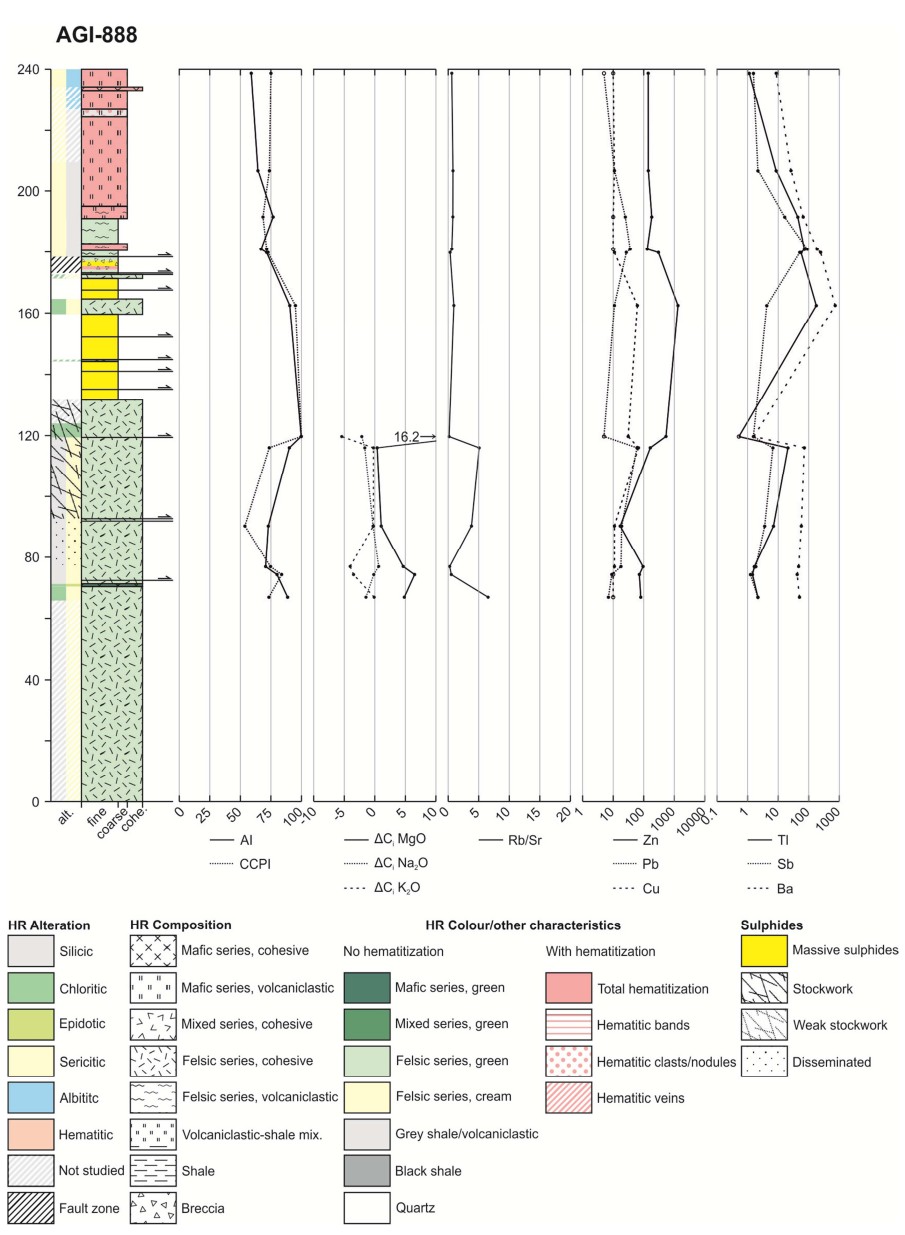


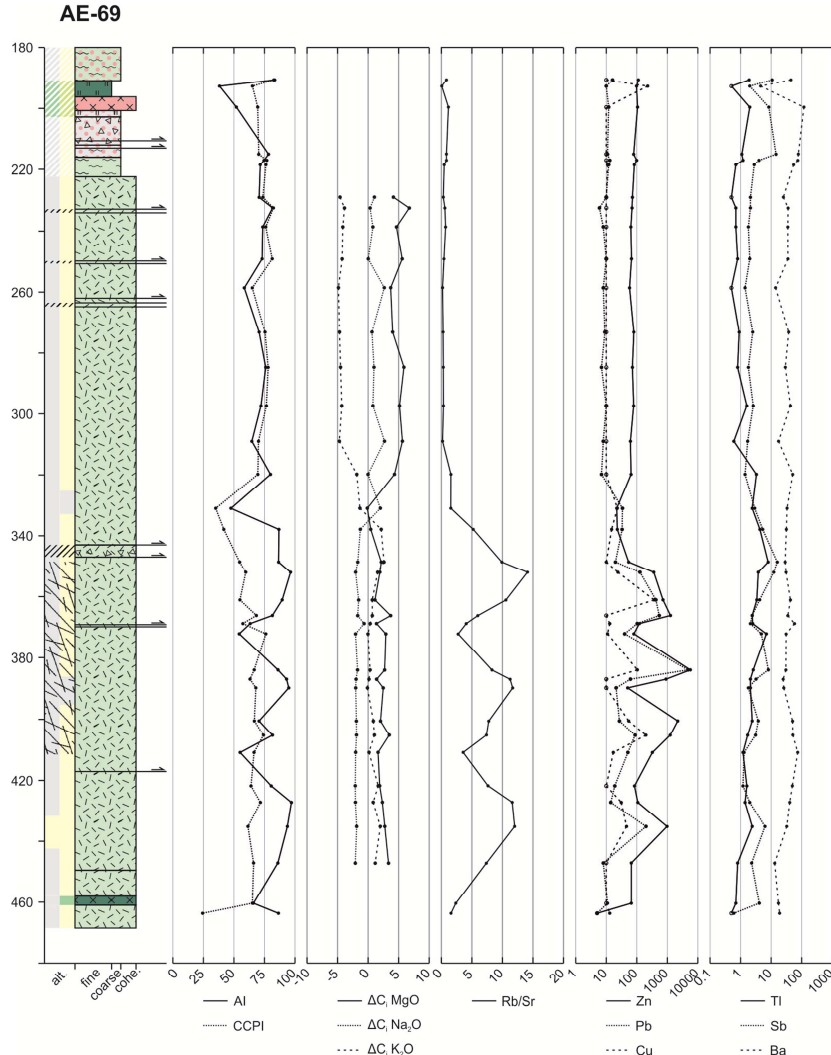

**Figure 11: Diagrams showing the lithology and alteration assemblages and whole rock geochemistry of analysed samples along drill cores AGI-888 and AE-69. AI, CCPI in %; $\Delta C_i$ MgO, $\Delta C_i$ Na$_2$O and $\Delta C_i$ K$_2$O in wt. %; trace element contents in $\mu$gg$^{-1}$.**

Changes within and close to the stockwork zone in this distal area reflect the influence of deep hydrothermal fluids. In the most distal AE-68 stockwork zone there is a slight depletion in Na$_2$O associated to MgO enrichment, with no variation in K$_2$O. In the slightly more proximal AE-69, Na$_2$O depletion is coupled with a slight MgO depletion and significant K$_2$O enrichment, which are likely related to higher hydrothermally controlled sericitization; additional K$_2$O may derive from the leached inner parts of the system.

In more proximal locations, ATS-7 shows a more chaotic behaviour, likely due to the nearly parallel character of the drill core relative to the margin of the stockwork system. On the other hand, rocks in the chlorite-rich inner stockwork alteration zone are markedly enriched in FeO, with less enrichment in MgO (e.g. samples in DST-332, deep stockwork); these are also coupled with marked Na$_2$O and K$_2$O depletion. Thus, there is a general FeO enrichment and alkalis depletion towards the centre of the hydrothermal system, with MgO showing less systematic trends. Regarding alkalies, whereas even in the most distal samples from the stockwork (AE-68) Na$_2$O is still leached from the host rock, K$_2$O leached from the inner portions of





the system seems to be released from the fluid in more proximal locations (e.g. stockwork in AE-69) producing a K₂O-richer halo within the external part of stockwork system, and with no K₂O content modification beyond this distance. A Na₂O richer zone around the hydrothermal system where the Na₂O leached from the central parts precipitates is inferred but was not identified in this study. In the footwall it may occur beyond the distal samples in AE-68; in the hanging wall isocon calculations could not be performed due to the heterogeneous lithologies. It is suggested that an important proportion of the

leached Na₂O may have been precipitated in the hanging wall, and involved in the formation of its sericitic (Na-rich micas are usually found in the hanging wall to VMS deposits, e.g. Soltani-Dehnavi et al., 2018a) and albitic alteration zones.

Remarkably, the highest MgO gains are associated to chlorite-rich veins/bands within less chloritic alteration zones (e.g. sericite-quartz alteration zone), such as at ca. 120 m in AGI-888 (proximal shallow stockwork) or ca. 500 m in AE-68 (distal

stockwork). These rocks show marked MgO but negligible FeO enrichment coupled with depletion in Na₂O and K₂O. We preliminarily interpret this behaviour as due to local circulation of hydrothermal fluids with higher sea-water content (high Mg, low Fe) compared to deep hydrothermal fluids.

Pearce element ratios (Pearce, 1968) using Al as immobile element were also used to study mass changes in the system.

Results were similar to those from the isocon method and are therefore not described.

Our results are broadly consistent with those obtained from a broader but less systematic sampling of the hydrothermally altered halo around the eastern zone of the Aguas Teñidas deposit by Bobrowick (1995), except for the depletion in Si in the inner system described by the author, which we have not observed. According to Bobrowick (1995) there are: 1) large

additions of Si + Fe + Mg in the quartz-chlorite central zone of the hydrothermal system; 2) Fe and Mg gain coupled with Ca + Na + Si ± K depletion in the inner chloritic zone; 3) Mg gain and Ca + Na + Si ± K loss in the strong sericite-quartz alteration; 4) low Si + Na + Ca gain, and Fe + Mg + K loss in the moderate sericite-quartz alteration; and 5) moderate Si + Na gain and general Ca + Fe + Mg + K loss in the weak sericite-quartz alteration zone. Bobrowick (1995) suggested that the observed Si trend resulted from an initial Si-undersaturated character of the upwelling hydrothermal fluids, which produced

Si leaching from the centre of the system and subsequent precipitation in the external parts leading to extensive silicification. The silicified core of the system is interpreted as having formed at a later stage due to a change in fluid conditions.

Chemical trends described in the Aguas Teñidas system are consistent with those reported from other VMS deposits in the IPB (e.g. Relvas, 1990; Madesiky and Stanley, 1993; Costa, 1996; Almodóvar et al., 1998; Sánchez-España et al., 2000;

McKee, 2003; Barret et al., 2008; Conde, 2016). At Rio Tinto, exposure characteristics allowed Madesiky and Stanley (1993) to perform a systematic sampling across the hydrothermal system, and to study the effects of hydrothermal alteration on two felsic and one mafic massive (flows/sills) volcanic units. Analysis based on Pearce Element Ratios (PER) showed an overall enrichment in Fe, Mg and Mn in the feeder zone and up to 1.5 km from its centre, and a depletion in Ca, Na and K detectable up to 2.5 km from the centre of the feeder system (Madesiky and Stanley, 1993). The Alteration Index of

Ishikawa et al. (1976) presented a positive anomaly similar in extent to the alkalies depletion zone. In another study of Rio Tinto deposit Costa (1996) reported equivalent results, and a similar element behaviour has also been observed at other deposits within the IPB (e.g. Sánchez-España et al., 2000; McKee, 2003). These trends are equivalent to those observed in VMS systems elsewhere (e.g. Large et al., 2001b; Barret et al., 2005), which are dominated by Fe enrichment in the core of the upflow zone, K₂O and MgO addition at the margins of the feeding system, and a general loss of CaO and Na₂O. They

have been related to the formation of Fe- and Mg-chlorite and muscovite and the Na- and Ca-bearing plagioclase breakdown, respectively (Hannington, 2014).





In terms of vectors to ore, in the Aljustrel area, Relvas et al. (1990) suggested that a diagram with $MgO/Al_2O_3$, versus $Na_2O/(Na_2O+K_2O)$, versus $(K_2O+BaO)/Al_2O_3$ shows the direction of decreasing distance to ore. These ratios illustrate the

nearly complete removal of alkalis in the core of the hydrothermal system, followed by a potassic (±Ba) zone (sericite-rich alteration zone), and finally grading into an ultraperipheral external zone where Na is fixed in alteration minerals. Other elemental ratios have also been used as vectoring tools elsewhere, such as $S/Na_2O$ in the Lens K of Rosebery deposit (Large et al., 2001b). This ratio varies over 5 orders of magnitude, increasing towards the S-rich, Na-depleted centre of the system. The authors describe it as a good vector within the footwall with sericite and pyrite-bearing alteration, and for up to 50 m

into the hanging wall. In weakly developed hanging-wall alteration its usefulness is seen to decrease due to the lack of significant sulphide development or albite destruction. According to Large et al. (2001b) $S/Na_2O$ allows tracking the ore stratigraphic position at locations up to several hundred metres from the ore at this site. The authors note, though, that the $S/Na_2O$ ratio may also be higher in pyritic shales that are unrelated to mineralization. Behaviour of this ratio at Aguas Teñidas seems to be similar, with an increase in $S/Na_2O$ towards the ore or centre of the stockwork, but is far from

systematic.

### 4.2.3 Vectors based on trace elements

Vectoring tools based on trace elements usually focus on the detection and study of geochemically anomalous halos of hydrothermally related elements around and away from deposits (e.g. Large et al., 2001b; Ames et al., 2016; Mukherjee and Large, 2017). Differences exist in the size of these halos depending on the permeability of the medium and the behaviour of

trace elements (Large et al., 2001a, Hannington, 2014). Base metals (e.g. Zn, Pb, Cu) typically produce restricted halos which extend at most some tens to few hundreds of meters away from the deposit, whereas the more easily transported volatile elements (e.g. Tl, Sb, Hg) may produce halos which may extend several hundreds of meters and are therefore amongst the most investigated fluid-mobile elements in VMS systems (Large et al., 2001a; Gibson et al., 2007, Soltani-Dehnavi et al., 2018). The origin of these halos is still under discussion, as both primary and secondary origins have been

proposed (e.g. Germann et al., 2003, Ames et al., 2016).

Geochemical characterization of rocks in proximal (e.g. AGI-808, AGI-888), medial (e.g. AE-68, AE-69) and distal (e.g. MU-2, MU-5) locations to the orebody has allowed us to establish the trace element background compositions of rocks in the area, and thus to detect changes produced by mineralizing hydrothermal fluids. Generic threshold values have been

established above which mineralizing fluids are most likely to have influenced a given rock composition based on our own data from Aguas Teñidas and literature data from sedimentary and volcanic rocks of the VSC and PQ groups in Aguas Teñidas and surrounding areas (e.g. Conde and Tornos, 2019) and other zones of the Iberian Pyrite Belt (e.g. Sánchez España, 2000). Consistently to existing literature, Zn, Pb, Cu, As, Cd, Sb and Tl have been found to be the most useful indicator elements. The proposed threshold values for the Aguas Teñidas area are: 200 ppm for Zn, 50 ppm for Pb, 150 ppm

for Cu, 75 ppm for As (not valid for black shales, which can have higher values unrelated to mineralization), 0.5 ppm for Cd, 15 ppm for Sb and 2.5 ppm for Tl. However, caution is required in the use of Cd, Sb and Tl threshold values because only data from this study are available.

Downhole plots provide further information on the behaviour of these elements (Fig 11 and Supplementary Material 1.4).

Background values occur in distal cores (MU-2 and MU-5), upper sections of more proximal AE-68 and AE-69 cores, and remarkably also along most of ATS-7, with < 20 ppm Cu and Pb, < 100 ppm Zn, around 1 ppm Tl, and < 5-10 ppm Sb. A minor high in Pb occurs related to red shales in MU-5 immediately on top of the host unit, whose origin is uncertain. In ATS-7, despite its proximity to the stockwork, no anomaly is detected except for a minor local high at ca. 60 m depth; this



indicates the low permeability of the host dome away from the stockwork and/or low propagation capacity of the studied
elements along this particular lithology.

In AE-69 high base metal contents occur within the distal stockwork, whereas Tl and Sb contents are higher by the margins
of this system. In the more distal stockwork in AE-68 no base metal anomalies have been detected, whereas slight Tl and Sb
ones occur related to the stockwork.


AM-49 intersects the margin of the massive sulphides by its western end, cutting only ca. 10 cm of massive sulphides with
no associated stockwork. Enriched Pb, Zn and Tl contents occur in the uppermost part of URD host dome immediately
below the massive sulphides, with concentrations decreasing downwards into the footwall for at least 30 m. Chemical trends
in the hanging wall are less clear. Zn content decreases abruptly into the hanging wall, whereas Tl anomaly seems to show a
decreasing trend with a local low in the cohesive lava above the massive sulphides. Cu shows a high in this same lava, likely
due to its original mafic composition.

AGI-808 and AGI-888 intersect thicker massive sulphides and the underlying external zone of the stockwork in a more
central location. Both cores show a geochemically anomalous envelope around the massive sulphides for base metals and
volatile elements which in AGI-888 peaks for most elements at the sill in the middle of the massive sulphides. Contrary to
AM-49, Pb and Zn anomalous concentrations extend also into the hanging wall. Zn presents a larger enrichment and a better
defined geochemical anomaly around the orebody compared to Cu and Pb. Differently from other cores, in these proximal
cores Ba also generates a significant anomaly, with a behaviour that mimics that of Tl. Sb and Tl anomalies extend beyond
the limit of those of base metals and Ba, as expected from their higher mobility, especially in the hanging wall, where
positive anomalies occur up to over 50 m above the massive sulphides. It is noteworthy that the chloritic band/vein at ca. 120
m depth shows a marked depletion in Tl, Sb, Ba and Pb relative to the general trends of the respective halos, whereas Zn
content presents no apparent disruption.

Previous observations provide information on the behaviour of indicative trace elements, and on the chronology and genesis
of the geochemically anomalous halos as well as of other structures such as hanging wall shear zones or the chlorite-rich
band in AGI-888. Regarding element behaviour, our observations are consistent with the known mobility of the studied
elements in VMS-related hydrothermal systems (Large et al., 2001a; Gibson et al., 2007), with Cu producing the most
proximal anomalies, followed by Zn and Pb, then Sb, and finally Tl. This behaviour can be seen within the stockwork
system, as described in the distal stockwork in AE-69 and AE-68. There, in the slightly more proximal AE-69, base metal
anomalies occur within the stockwork and Tl and Sb ones outside it, whereas in the more distal AE-68 no base metal
anomalous contents occur and, instead, the anomaly is restricted to within the stockwork and marked by Tl and Sb. An
equivalent behaviour can be expected in the stockwork located more proximal to the orebody. Tl and Sb positive anomalies
are seen in the external stockwork in AGI-888, whereas we suggest that more central zones of the stockwork below the
massive sulphides (e.g. chloritic alteration zone) should be depleted in these elements. Although no chemical profiles have
been analysed for the central parts of the stockwork (e.g. along DST-332), the two samples analysed from the chloritic zone
in DST-332 (DST-332/252.5 and DST-332/275.9) show low Tl (<detection limit and 1.2 ppm, respectively) and Sb (2.8 and
1.4 ppm, respectively), supporting our hypothesis. This behaviour is also observed outside the stockwork, as in the hanging
wall. In the latter case, our data also show the restrictions on halo formation by host rock permeability, with minor
propagation into massive facies (e.g. in AM-49, ATS-7 and AE-69), as seen in other study areas (e.g. Large et al., 2001a,
Ames et al., 2016). Finally, there are differences in the extent and composition of the geochemically anomalous halo along
the mineralized system, which are likely related to the intensity and characteristics of the hydrothermal fluid circulation. In





marginal locations of the orebody (e.g. AM-49), it extends to shorter distances compared to more central locations (e.g. AGI-888), especially towards the hanging wall. Moreover, in central locations an additional indicator element is found (Ba), which does not occur in other areas. Thus, Ba may be a good vector towards the central area of the system.


The chlorite-rich vein at ca. 120 m in AGI-888 strongly contrasts with its host rock in terms of mineralogy and whole rock composition of both major and trace elements. The host rock presents sericite-quartz alteration and follows the regular trend of the base metal and volatile elements halo around the massive sulphide. In contrast, this vein shows pervasive chlorite alteration, very high Mg enrichment, and depletion of indicative elements defining the geochemical halo around the massive

sulphide orebody except for Zn. Chloritization associated to hydrothermal fluids related to the mineralizing process produces enrichment in Fe supplied by the fluid, as seen in the central deep stockwork in DST-332. In this vein, though, the lack of Fe enrichment indicates a different fluid composition, while exceptional Mg enrichment suggests that it was seawater-dominated. Regarding its relative chronology, negative anomalies in halo-related elements suggest that fluid circulation was active at least during and/or after the formation of the geochemical halo. Moreover, presence of Zn contents consistent with

the geochemical halo suggests that the halo was partially established when the chlorite vein system formed, and that circulating seawater-dominated fluids subsequently remobilized and leached all indicative elements except for Zn. The different behaviour of Zn may provide information on the circulating fluid properties, whose investigation is beyond the scope of this work. Sub-seafloor shallow mixing of external seawater with deeper hydrothermal fluids is regarded as a common process controlling the behaviour and formation of modern and ancient VMS systems, and is suggested to be driven

by the convection triggered by the ascent of hot deep hydrothermal fluids (Large et al., 2001a, Franklin et al., 2005; Tornos, 2006; Tornos et al., 2015). Although the introduction of external seawater into the underground hydrothermal system is usually considered to occur through diffuse percolation, here we suggest that the aforementioned vein could represent a focused feeder zone, maybe produced by concentration of the originally diffuse flow along a favourable structure.

Geochemical halos also provide information on the chronological relationship between the orebody and the hanging wall sequence, as well as on shear zones on top of the massive sulphides. Given the replacive character of Aguas Teñidas deposit, the chronologic relationship between the massive sulphides and the immediate hanging wall materials is unknown. Two main scenarios can be considered: 1) The geochemical halo formed after tectonic deformation, long after the mineralizing event, in which case the halo would be produced through element remobilization by metamorphic fluids; or 2) The halo

formed prior to tectonic deformation. The presence of hydrothermal alteration in the hanging wall to the Aguas Teñidas deposit with characteristics consistent with alteration observed in other districts and clearly associated to VMS deposits formation is a strong argument in favour of hanging wall hydrothermal alteration and halo formation contemporary or soon after the main orebody formation, and controlled by related hydrothermal fluids either during peak activity or the waning stage. Additionally, if the interpretation of the chlorite vein in AGI-888 as a seawater feeder zone holds true, it would be an

argument against the post-deformation and metamorphic-fluid controlled formation of the geochemical halo. On the other hand, the thrusted character of the hanging wall over the orebody is not necessarily an argument against the genesis of hydrothermal alteration and geochemical halo prior to tectonic deformation. The stratigraphic sequence at Aguas Teñidas is rich in faults and shear zones of unknown displacement. Most of these faults occur within given lithological units (e.g. the host dome or even the massive sulphides), which indicates minor displacements. Thus, it is considered that tectonic

deformation at Aguas Teñidas likely involved the stacking of many minor structures with small displacement producing an overall "shear-like" deformation, rather than fewer structures with larger displacements. Similar tectonic configurations have been described elsewhere in the IPB, such as at the Puebla the Guzmán Antiform (Mantero et al., 2011). If this holds true, the studied hanging wall to the massive sulphides could be proximal to its original position, and thus the geochemical halo could be of primary mineralization-related origin and/or related to early remobilization (prior to tectonic deformation).




Finally, indicative trace elements have been used to investigate the location of the seafloor contemporary to the formation of the Aguas Teñidas orebody within the stratigraphic sequence. Since no potentially syn-mineralization exhalative deposits have been described to date, we searched geochemically anomalous stratigraphic horizons which could indicate exhalation of hydrothermal fluids into the sea bottom. Volcaniclastic materials immediately above the host dome away from the massive

sulphides and associated geochemical halo (e.g. in AE-68, AE-69) show no geochemical anomaly which could indicate a preferential circulation of the hydrothermal fluids along these horizons to significant distances, or deposition of these materials within a sea-bottom water mass chemically modified by upwelling hydrothermal fluids. The sampling above this immediate hanging wall is not yet detailed enough to locate the potential seafloor, and thus more work is still needed to solve this question.


The indicator elements here described have been found to be the most useful single-element whole rock geochemistry pathfinders to ore in the Aguas Teñidas system, and we have shown that they represent powerful tools not only for orebody vectoring, but also for the investigation and understanding of mechanisms and chronology controlling the formation of this VMS deposit.


Detailed descriptions on indicator trace elements such as the one presented here are lacking in other deposits of the IPB. However, more general trace element behaviour trends have been studied at other deposits such as Rio Tinto (Piantone et al., 1993, 1994; Costa, 1996), Aljustrel (Barriga, 1983, Relvas, 1991) or Masa Valverde (Toscano et al., 1993). Similar to Aguas Teñidas, high Tl, Se, Sb and Hg, and relatively high Zn and As contents have been described in halos proximal to the

mineralization (e.g. Piantone et al., 1993). A strong correlation between Tl and Ba was also observed at Rio Tinto by Costa (1996); these elements are enriched in the sericite alteration zone together with Se and Sb, forming a proximal geochemical anomaly (up to 500 m) relative to the centre of the hydrothermal system (Costa, 1996). Ba enrichment in sericitic alteration also occurs at Aljustrel (Barriga, 1983, Relvas, 1991; Barret et al., 2008) and Masa Valverde (Toscano et al., 1993). At Feitais orebody, in Aljustrel deposit, the Ba enrichment halo has dimensions of up to tens of metres above and laterally from

the orebody, similar to those observed at Aguas Teñidas (Barret et al., 2008).

Geochemical halos of volatile elements are a commonly used vectoring tool in other VMS districts (e.g. Mount Read Volcanics, Australia, Large et al., 2001b; Flin Flon Mining Camp, Canada, Ames et al., 2016). The comparison of several VMS systems in Australia by Large et al. (2001a) revealed that there is a relationship between the Zn/Cu content of the

deposits and the extent of the volatile elements halo; Zn-rich deposits (e.g. Rosebery, Hellyer, Thalanga) present larger Tl and Sb halos, whereas in the Cu-Au type deposits (e.g. Western Tharsis, Highway-Reward) these are more restricted. Rosebery and Hellyer have halos in which Tl and Sb concentrations higher than 1 ppm can be found up to several hundred meters away from the deposit. In contrast, in Thalanga this halo extends less than 50 m into the hanging wall and footwall (Large et al., 2001a). Although analysed less frequently, Hg has also been used as a vectoring tool, for example in the

Noranda district and Bathurst Mining Camp (Canada) (Boldy, 1979; Lenz and Goodfellow, 1993). Hg behaves similar to Tl and Sb, producing geochemical halos that are more developed in the hanging wall to the deposits (Gibson et al., 2007).

VMS deposits in the IPB fall closer to the characteristics of Zn-rich deposits described by Large et al. (2001a), like Rosebery deposit, which can provide a reference for the geometry of original halos in less tectonically deformed areas. In Lens K in

Rosebery deposit (Mount Read Volcanics, Tasmania, Large et al., 2001b), the Zn and Pb halo reaches about 400 m along the ore stratigraphic horizon and 20 to 50 m across stratigraphy into the footwall. Cu, which usually requires higher transport temperatures (Hannington, 2014), produced a significantly smaller halo. In contrast, Tl defines a halo at least 270 m across



stratigraphy around the ore position and, opposite to most vectoring tools, is better developed in the hanging wall than in the footwall. In the hanging wall Tl concentrations are higher than 1 ppm for over 200 m; in the footwall Tl halo extends at least

70 meters below the ore lens. At Rosebery Sb varies less systematically than Tl, and is therefore recommended to be used in combination with the latter by Large et al. (2001b). The lateral extent of Tl and Sb halos along the ore stratigraphic horizon extends over 500 m, beyond the limits of sampling (Large et al., 2001b).

In addition to single elements, trace element ratios have also been used as vectoring tools. For example, in the Lens K of

Rosebery deposit Large et al. (2001b) use the Ba/Sr ratio. This ratio increases towards the ore as a result of Ba substitution for K in white mica, and to Sr depletion due to albite destruction. The authors detect a broad halo of higher Ba/Sr that extends up to 80 m into the hanging wall sequence. They consider that this ratio is superior to most other indexes (but not Tl) in defining halos in the hanging wall directly above the ore and for some distance lateral to ore, but that it becomes a less distinct vector at distal positions from the ore. Large et al. (2001b) also investigated the usefulness of the Rb/Sr ratio, which

has a similar pattern to the Ba/Sr ratio but is less anomalous in the hanging-wall sequence.

### 4.3 Portable XRF

The vectoring tools based on whole rock geochemistry described in previous sections traditionally rely on conventional laboratory-based XRF and ICP analysis. These techniques provide superior accuracy and precision. However, they are coupled with high costs - related to both sample preparation and analysis - which usually result in low spatial resolution in

systematic studies related to exploration, long time lapse between sample collection and analytical results, and a destructive character of sample preparation. These aspects reduce the efficiency of these methods for obtaining whole rock geochemistry data during active exploration. Thus, efforts have been devoted to implementing the use of portable XRF as a fast, cost-effective first-stage tool in lithogeochemical exploration in VMS and other mineral systems (e.g. VMS systems, Ross et al., 2014, 2016; McNulty et al., 2018, 2020; komatiite-hosted nickel sulphide deposits, Le Vaillant et al. 2014; Au in greenstone

belts, Glazley et al., 2011; laterites, Duee et al., 2019), analysing varied materials such as rocks, soils or tills (e.g. Hall et al., 2016). P-XRF can be used to obtain geochemical data faster and/or to fill the gaps between traditional laboratory analyses.

Taking into account these considerations, we have tested if the lithogeochemical vectors described in previous sections are detectable and usable through direct analysis of core samples with p-XRF. Analysis on unprocessed rock samples using a

single analysis mode (Cu/Zn mining mode) and low counting times (30 s per beam for a total of 120 s per analysis) was chosen as we consider that this represents realistic and convenient analytical conditions under which exploration work can be carried out (further discussion is available in Supplementary Material 1.3).

### 4.3.1 Lithological unit recognition

P-XRF analysis has been shown to be useful in VMS systems for discrimination of lithological units (e.g. using Ti/Zr, Al/Zr

and Zr/Y ratios, Ross et al., 2014, 2016). At Aguas Teñidas the most useful diagram for lithological and unit discrimination is Ti/Al$_2$O$_3$ vs Zr/Al$_2$O$_3$ (Fig 8a). Figure 12 compares laboratory and p-XRF results for 5 samples representative of the whole compositional range of the host sequence. For each sample, the data obtained from 1) laboratory analysis (solid symbols), 2) pressed powder pellet p-XRF analysis (faded symbols), and 3) average hand specimen p-XRF analysis (yellow pentagons) are represented. In addition, for samples AM-49/879.75 and AE-68/101.8, the envelopes containing single spot-analyses, 7-

point averages, and 10-point averages for hand specimen analyses are also shown to provide a general idea of the effect of analytical precision and multiple-spot averaging.



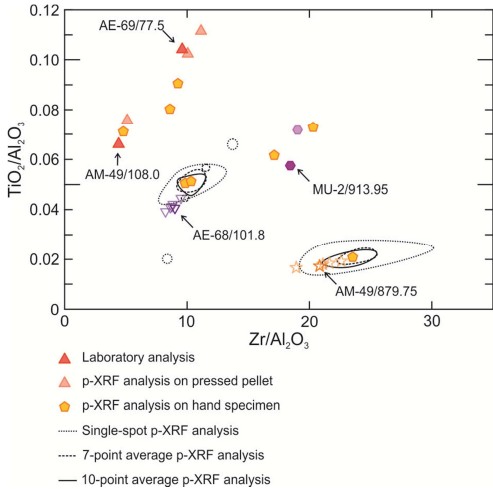

**Figure 12: TiO$_2$/Al$_2$O$_3$ v. Zr/Al$_2$O$_3$ discrimination diagram comparing whole rock geochemistry data obtained from laboratory and**
**p-XRF analysis of representative samples. Ratios calculated with TiO$_2$ and Al$_2$O$_3$ contents in wt % and Zr contents in μgg$^{-1}$.**

Pressed powder pellet data occur close to the laboratory data, along trends departing from the axis origin which are likely
due to lower precision in Al measurement compared to Ti and Zr. The same effect can be observed on the envelopes for
samples AM-49/879.75 and AE-68/101.8. Data from sample AE-68/101.8 stress the importance of averaging enough spots
for the lithogeochemical characterization of heterogeneous rocks, as the effect of chemical outliers is reduced with the
number of averaged spots. For AM-49/879.75 (average of 21 spots), AE-101.8 (average of 21 spots) and AM-49-108.0
(average of 7 spots), hand specimen results show higher TiO$_2$/Al$_2$O$_3$ and Zr/Al$_2$O$_3$ compared to laboratory and pressed pellets
data. This is likely due to a differential decrease in the signal of Al on one side, and Ti and Zr on the other, caused by sample
surface roughness and irregularities (Duee et al., 2019; Supplementary Material 1.3). In sample AE-68/101.8 two sets of 21
analyses were performed, one on the external curved rough surface of the drill core and another on the flat cut surface
obtained from sawing the core in two. Average compositions are nearly indistinguishable, thus showing that the analysis of
the external part of cores is a valid approach, and that flat surfaces are not needed. Results from hand specimen analysis of
samples AE-69/77.5 and MU-2/913.95 show the importance of monitoring equipment drift during measuring sessions. Due
to an unknown error likely related to sensor heating, during the measurement of these two samples light element
concentrations –especially Al and Mg - shifted towards unreasonably high values. This resulted in erroneously decreased
TiO$_2$/Al$_2$O$_3$ and Zr/Al$_2$O$_3$ ratios for one hand specimen measurement of MU-2/913.95, and for both of AE-69/77.5 (cut
surface and core surface).

Figure 12 shows that the highest uncertainties in p-XRF geochemical characterization of both pressed pellets and hand
specimens are due to the lower precision in light elements determination, particularly Al, and to the higher influence of
factors such as surface roughness, distance between sample and detector, or moisture, on the measured intensities for these
elements (discussed in more detail in Supplementary Material 1.3). Thus, the use of other diagrams avoiding light elements
is strongly recommended when working with p-XRF data. The Nb/TiO$_2$ vs. Zr/TiO$_2$ diagram in Figure 13 provides an
example. Although less efficient at discriminating variations within the mantle (data tightly grouped at low values for both
ratios) or mixing series (no discrimination between mixing trends of components with different Zr/Al$_2$O$_3$ ratios), this
diagram effectively separates mantle, mixing and crustal series, different compositions within the last class (large
compositional range at high values for both ratios), and the presence of a sedimentary component, which produces a depart
from the igneous trends towards higher Nb/TiO$_2$ values.





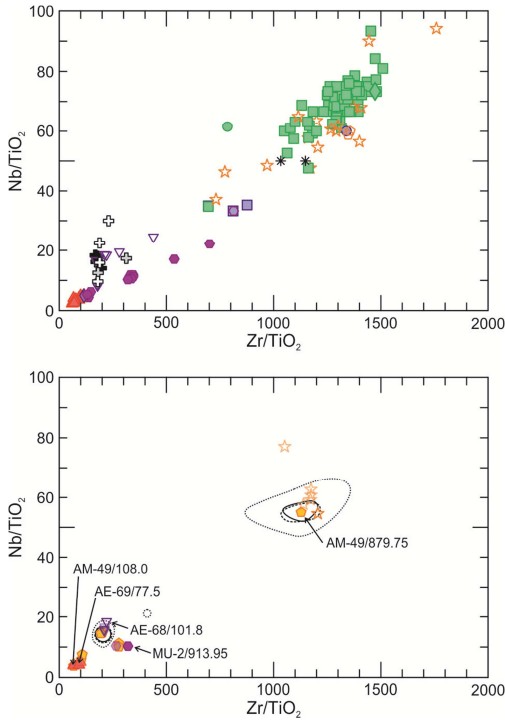

**Figure 13: Nb/TiO₂ v. Zr/TiO₂ discrimination diagrams showing (a) laboratory whole rock geochemistry data and (b) comparison between laboratory and p-XRF whole rock geochemistry data of representative samples. Ratios calculated with TiO₂ content in wt % and Nb and Zr contents in μgg⁻¹. Symbols follow Figure 7 in (a) and Figure 11 in (b).**

### 4.3.2 Major elements

Na₂O and MgO contents, on its own or in ratios (e.g. Alteration index), have been shown to be useful indicators of alteration and vectors within the hydrothermal system of VMS deposits in the previous section. However, Na₂O cannot be measured by p-XRF devices, and MgO typically presents low precisions. Therefore, alternative ratios using more robust elements have been proposed to track hydrothermal alteration during exploration by p-XRF in VMS system. For instance, the Rb/Sr ratio is suggested to approximate the behaviour of the Alteration Index (McNulty et al., 2020). At Myra Falls, Rb/Sr ratios vary

from <0.1 for least altered rocks, 0.1 to 0.5 for weakly altered rocks, 0.5 to 1.0 for moderately altered rocks, 1.0 to 2.0 for strongly altered rocks, and >2.0 for intensely altered rocks (McNulty et al., 2020). Our laboratory data confirm this similarity (Fig. 11 and Supplementary Material 1.4), which is most evident in AE-69, although differences exist. For example, in AGI-888, whereas AI depicts a nearly symmetrical pattern around massive sulphides, higher Rb/Sr values only occur within the footwall to the massive sulphides. Therefore, caution is advised in the use of this ratio. Additionally, it is noted that the

chlorite band around 120 m depth shows a low in Rb/Sr as is also the case with other indicators such as Tl, Ba or Pb, further confirming its singular origin. Figure 14 shows that Rb/Sr measured by p-XRF on hand specimen (both from core surface and cut sections) are equivalent to ratios measured on pressed pellets and in laboratory. Thus, Rb/Sr can be confidently measured using p-XRF devices under the conditions used in this study.



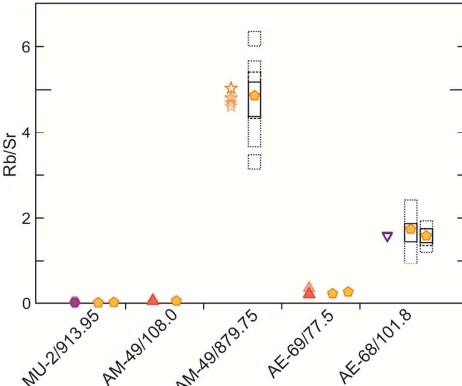

**Figure 14: Comparison between Rb/Sr data obtained from laboratory and p-XRF whole rock geochemistry analyses of representative samples. Symbols follow Figure 11.**

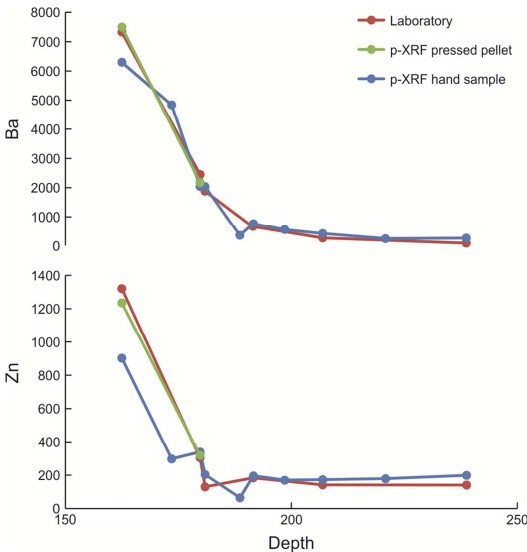

**Figure 15: Ba and Zn geochemical profiles along the hanging wall in core AGI-888. Depth in m along core; Ba and Zn concentrations in in μgg[-1].**

**4.3.3 Trace elements**

To test the usefulness of p-XRF in the analysis of trace elements used in ore exploration, the geochemical halo in the hanging wall to the massive sulphides in core AGI-888 was studied. Figure 15 shows Ba and Zn chemical profiles along the core obtained from the analysis of samples in the laboratory and by p-XRF on hand specimen (10 points average) and pressed pellets (5 points average). More samples were analysed by p-XRF than in the laboratory, showing the usefulness of

this tool to fill in the gaps between traditional laboratory data. Ba p-XRF profile closely matches laboratory data, whereas Zn data show values lower than expected in some hand specimens likely related to the nugget effect. Within the VMS alteration halo Ba is typically hosted in white mica (Large et al., 2001b; Soltani Dehnavi et al., 2018a), which tends to occur pervasively throughout rocks (Large et al., 2001a; Franklin et al., 2005; Soltani Dehnavi et al., 2018a); on the other hand, Zn tends to occur as sulphides, which may present a more heterogeneous distribution either as disseminate or small veins in

lightly mineralized samples, thus presenting a higher nugget effect (Bourke and Ross, 2015). This behaviour stresses the importance of averaging enough analyses per sample. Whereas 3 points may be enough for Ba characterization, it may produce too high deviations for elements with behaviours similar to Zn. Other useful elements such as Tl and Sb are also




typically hosted in white micas, thus presenting a distribution similar to that of Ba (Soltani Dehnavi et al., 2018a), although Tl contents are usually too low for precise p-XRF determination.

**5. Summary and conclusions**

In this work we have studied and characterized vectors to ore related to mineralogical zoning and whole rock geochemistry in the case study of the Aguas Teñidas replacive volcanic rock hosted VMS deposit of the IPB.

Alteration halos around the main orebody and associated stockwork show a geometry and distribution equivalent to other
VMS deposits in the IPB and other districts (Fig. 16). In the footwall, a concentric cone-shaped hydrothermal alteration which bears the stockwork passes laterally, from core to edge, from quartz (only locally), to chlorite, sericite–chlorite, and sericite alteration zones, all of them with quartz as an alteration phase. In the hanging wall, hydrothermal alteration occurs despite its thrusted character; a proximal sericite alteration zone is followed by a more distal albite one, which is described here for the first time in the IPB.


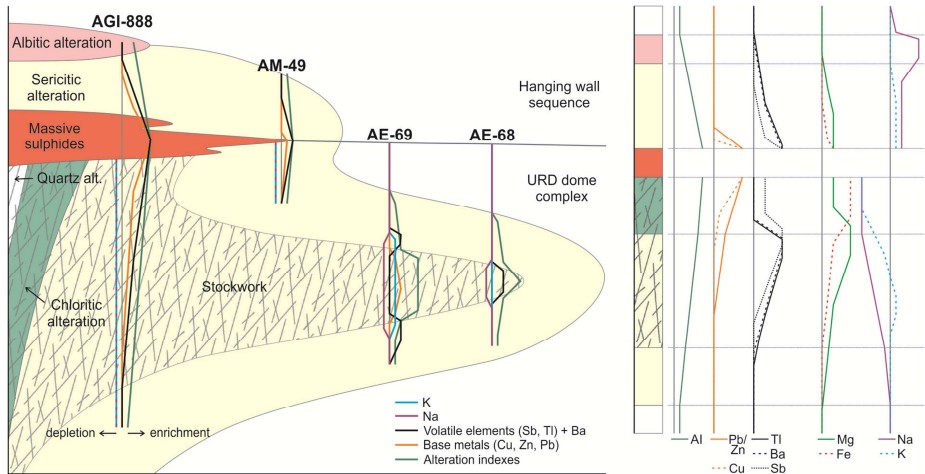

**Figure 16: Schematic summary of the main observations performed in this study on vectoring tools related to hydrothermal alteration mineral zoning and whole rock major and trace elements. Sodium enrichment in the albite alteration zone is inferred and needs to be confirmed. Not to scale. AI: Alteration indexes.**


The lithological units in the host sequence have been characterized using whole rock geochemistry. Discriminant diagrams based on immobile element ratios have been elaborated ($TiO_2/Al_2O_3$ v. $Zr/Al_2O_3$, $Nb/Zr$ v. $TiO_2/Al_2O_3$) which allow identifying specific units (e.g. the dacitic dome host to the massive sulphides), different magma series (e.g. mantle-derived, crustal-derived, mixing series), and sediment component. The identification of specific lithological units based on whole
rock geochemistry represents a powerful exploration tool in a heavily tectonized area such as the IPB.

Major element variations related to hydrothermal alteration useful for vectoring purposes have been studied through the investigation of alteration indexes and mass balance calculations using the isocon method (Fig. 16). Alteration and Chlorite-Carbonate-Pyrite Indexes generally increase towards the centre of the system, but caution is required in their use due to the
influence of the original rock composition and prior seafloor metasomatism on their final values. Regarding single element variations, only the compositionally homogeneous footwall host unit could be studied using the isocon method. There is a general FeO enrichment and alkalis depletion towards the centre of the hydrothermal system, with MgO showing less





systematic trends. Regarding alkalies, $K_2O$ leached from the inner portions of the system is released from the fluid in marginal to distal locations of the stockwork system, whereas even in the most distal analysed samples from the stockwork

$Na_2O$ is still leached from the host rock. A $Na_2O$ richer zone around the hydrothermal system where the $Na_2O$ leached from the central parts precipitates is inferred but has not been identified in this study.

Base metals (Zn, Pb, Cu), volatile elements (Tl, Sb) and Ba have been found to be the most useful trace elements for vectoring purposes at Aguas Teñidas. Consistently with the known mobility of the studied elements in VMS-related

hydrothermal systems, Cu produced the most proximal anomalies, followed by Zn and Pb, then Sb, and finally Tl and Ba (Fig. 16). At locations away from the massive sulphides, geochemical halos associated to these elements occur within the stockwork or immediately around it. At marginal zones of the massive sulphides, the halo is restricted to the footwall for base metals. In central parts of the deposit, a well-developed geochemical halo occurs in both footwall and hanging wall. Remarkably, a significant Ba halo has only been detected in this central area, indicating the importance of Ba as a vector

towards the central part of the hydrothermal system.

Additionally, threshold values have been defined for these trace elements which can be used in the area around Aguas Teñidas to identify rocks which are likely to have been affected by the mineralizing hydrothermal system. These are: 200 ppm for Zn, 50 ppm for Pb, 150 ppm for Cu, 75 ppm for As (not valid for black shales, which can have higher values

unrelated to mineralization), 0.5 ppm for Cd, 15 ppm for Sb and 2.5 ppm for Tl. However, caution is required in the use of Cd, Sb and Tl threshold values because only data from this study are available.

Trace elements have also provided valuable information related to the genesis and evolution of the deposit. A possible focused feeder zone of seawater into the shallow hydrothermal system has been identified, which was active during and/or

soon after formation of the orebody. Additionally, this possible feeder zone constrains the formation of the geochemically anomalous halo of the studied trace elements around the deposit to the period of deposit formation, thus precluding halo formation by subsequent remobilization by metamorphic fluids. Trace element halos also indicate that thrusts affecting the hanging wall to the deposit must have had minor displacements, which is consistent with the presence of hydrothermal alteration in this zone.


Finally, the usefulness of p-XRF devices for the analysis of previously described chemical vectors in realistic exploration conditions has been tested. Our results show that the proposed vectors, or adaptations designed to overcome p-XRF limitations, can be confidently used by analysing unprepared hand specimens, including the external rough curved surface of drill cores.


The results presented in this work contribute to the characterization and understanding of vectors to ore in replacive VMS systems of the IPB, thus improving mineral exploration and the location of new resources in the area. In addition, data will not only be applicable to exploration in the IPB, but on a broader scale will also contribute to improve our general understanding of vectors to ore in replacive-type VMS deposits elsewhere.

**Author contribution**

GG: conceptualization (equal), investigation (lead), writing – original draft (lead); FT: conceptualization (equal), funding acquisition (lead), writing – original draft (supporting); EL: investigation (supporting), writing – original draft (supporting);



JMP: investigation (supporting), writing – original draft (supporting), funding acquisition (supporting); JCV: investigation (supporting), writing – original draft (supporting), funding acquisition (supporting).

**Competing interests**

The authors declare that they have no conflict of interest.

**Acknowledgements**

The authors want to thank: MATSA for granting access to Aguas Teñidas drill cores, information supplied, and assistance during core investigation and sampling; technicians in the Laboratorio de Petrología y Geoquímica of the Universidad
Complutense de Madrid for assistance in sample processing; J. Montes for thin section preparation; M. Álvarez and R. Fort for granting access to their NITON XL3t 900Analyzer. This research has been conducted within the NEXT (New Exploration Technologies) project and has received funding by the European Union's Horizon 2020 research and innovation programme under Grant Agreement No. 776804.

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
