# Peer review of "Vectors to ore in replacive VMS deposits of the northern Iberian Pyrite Belt: mineral zoning, whole rock geochemistry, and application of portable X-ray fluorescence"

_Solid Earth, 2021_

## Referee Comment (RC1)

**Vectors to ore in replacive VMS deposits of the northern Iberian Pyrite Belt: mineral zoning, whole rock geochemistry, and use of portable XRF**

The paper entitled "Vectors to ore in replacive VMS deposits of the northern Iberian Pyrite Belt: mineral zoning, whole rock geochemistry, and use of portable XRF" presents mineralogical and bulk geochemical datasets from the Aguas Tenidas VMS deposit in the Iberian Pyrite Belt. The study focuses on the application of trace metal ratios and concentrations in bulk-rock powders to the discovery of VMS mineralisation. In addition to this the study compares data obtained from whole-rock (ICP + XRF) data with portable XRF analysis. The latter in my opinion is very important and gives the paper a clear application to the mining and mineral exploration sector. In parts I found the structure confusing and think it needs to be more clearly separated into proximal and distal facies in terms of both mineralogy and geochemistry. I struggled to link the core and core location to the geochemical data. To overcome this, I suggest including representative core photos and more logs like those included in Figure 11. Generally there was a lack of quantification throughout with terms such as "hot", "proximal", "slight" or "laboratory" but no explanation/quantification was given as to what these terms specifically meant. Clearly there is an interesting story here but the manuscript was very long and unclear in some areas. At the moment I think the key points are lost in amongst extraneous information which makes the paper confusing to read.

Please see my detailed comments below:

Line 1: Title, consider "replacement style" instead of "replacive"

Line 3:  Do not abbreviate XRF in the title, spell it out.

Line 9: I would not state mineral systems – I think ore deposits or economic mineralisation would be more appropriate. Mineral systems gives the impression of mineralogy or mineral-scale chemistry.

Line 10: Define what you mean by "vectors to ore" – geochemical, mineralogical, other or both?

Line 11: suggest swapping "outstanding" for "world-class"

Line 21: What does XRF stand for?

Line 31: I question this statement as the analytical spot size of a pXRF is a few microns (?). What if you have coarse mineral grains? The mineral scale partitioning of different elements between minerals will be important and severely bias your data.

General comment, throughout I suggest change "replacive" to replacement type.

Line 40: Under covering? Are you referring to increased overburden or simply that we are having to hunt in hard to reach places such as tundra and rainforests? Please clarify.

Figure 1: Nice figure, some labels such as CCPI need to be explained. What do the two black dashed lines represent?

Line 62: Give an approximate tonnage, see Galley et al., 2007 (I think their figure is >1.5 Bt of sulfide ore).

Line 83: What do you mean by mineralogical zoning? Alteration halo?

Line 92: Define "hot" and I suggest replacing with "high temperature" ~350°C

Line 92: ambient → surrounding

Line 93: feeder zone… that directly underlies the massive sulfide mound

Line 93: You keep mentioning replacement, you should add a sentence explaining exactly what you mean by this and how this differs from the typical exhalative VMS deposit model.

Line 103: I would separate this into hypogene mineralizing processes and then anything that is secondary in origin – why are the secondary processes important? Because they can modify the primary signatures.

Line 108: remove "related to"

Line 116: You state a tonnage for La Zarza why not for Neves Corvo etc.?

Line 119: Specify an age next to the geological time periods, this would be easier for the reader to understand.

Lien 130: oldest to youngest.

Line 133: Chemical sediments? Please specify a rock type.

Line 134: I would insert a reference back to Fig. 2 here as you are discussing the location of the deposit.

Line 141: very minor point… comma is missing after "et al." on Velasco-Acebes et al. 2019 (and some other reference throughout).

Line 148: Is there an approximate temperature range for this? Or at least specify the upper temperature limit if possible. Green schist should be one word.

Line 151: Use of "ores", are they actually ores, do they have an economic value?

Lien 151: This should be integrated with above paragraph or deleted – it seems a little out of place.

Line 155: "The" Agus Tenidas deposit

Line 160: Could you state an approximate depth of burial/overburden?

Line 161: Be consistent, is it W or west?

Line 176: (Figure 3) I would instead say aerial view or plan view and cross section view looking (X direction). You should also state what the black lines are, it's obvious to me that they are drill holes (I hope!) but this should be stated.

Line 183: Delete "at large"

Line 190: In what quantity are these minerals present? Presumably trace amounts <1%?

Line 193: Avoid 1 sentence paragraphs, integrate with above paragraph.

Line 197: Replace "material" with lithologies.

Line 201: Dismembered by tectonics? Do you mean faulting during later tectonic events?

Line 208: add "The" to the beginning of the sentence.

Line 213: What do you mean when saying "strong vertical and lateral facies changes"? Please specify, for example, in my mind I think of grain size or rock type, is that correct?

Table 1: Do not abbreviate unit to U in the table – if you write unit than you can delete the name you repeat in the brackets below and just include the abbreviation – this would make the table clearer.

In the table you use terms like "rich" and "less" can these be quantified?

Line 223: Other deposits, such as?

Line 227: Earlier in the MS (Line 148) you state up to greenschist facies metamorphism

Line 228: Remobilisation of what?

Line 230-231: I would deleted this, you state this is more detail in section 3.1, no need to repeat.

Line 239: Don't start a sentence/paragraph with an abbreviation, spell it out. When you say "whole rock analysis" what does this mean?

Line 242: Is there a document provided by SGS outlining the detailed methodology? If so, this should be reference here.

Table 2:

MU5 – define "close to MU-2"

In general I found the table hard to follow. This should be moved to the appendix, instead why not include some representative core photographs? I think it would be more useful for the reader to visualise what the core looks like. Core logs for key boreholes as well perhaps.

Figure 6: Referring back to Fig. 3 for the legend is not appropriate, this needs to be included in this figure.

Line 251: A selection of samples, n =?

Line 252: GOLDD Technology? I have no idea what this means, delete or specify.

Line 266: Avoid 1 sentence paragraphs... integrate with above section.

Line 273: You mention that it confirms their observations but you need to state what these observations are.

Line 276: What does distal mean? 10 m or 10 km, this is important and not clear to me.

Line 277: But the rocks surrounding the deposit have also been exposed to prehnite-pumpellyite facies metamorphism so they don't only preserve seafloor metasomatism.

Line 283: Change "mafic crystals" to igneous glass or groundmass.

Line 287: You say that sericite-quartz alteration dominates? Is this not a function of host rock, i.e., you might have a higher proportion of felsic lithologies in these drillcore samples?

Line 299: add zone after hydrothermal alteration.

Line 310: "Minor to rare", this is subjective and should be quantified – <5 vol.%?

Line 311 and 313: Specify units for the depth measurements, presumably m.

Figure 7: petrographic microscope picture → photomicrographs

Again add units after measurements (m) – nice images!

Line 334: reference to Fig 7H that shows Fe oxides (?)

To make section 3.1 easier to understand I suggest sub-dividing it into proximal and distal sections and hangingwall and footwall. It becomes confusing when swapping between different drill holes and you easily lose the reader.

Line 365: Yes, I agree that hyperspectral imaging is great for exploration but you focus on drillcore samples, how is this method applicable to your data/deposit? I would summarise your key alteration findings here (i.e. the expected spectral response of proximal vs. distal alteration) and relate this to hyperspectral imaging.

Line 370: Unit → rock, also extra e on halos.

Line 373: I don't think mineralization related lithologies is what you are identifying, rather it is chemical changes that related to specific hydrothermal/mineralising events.

Line 374 are "to ore deposits" after pathfinder

Figure 8: In the text you reference figure 8b – A and B are missing on the figure.

Figure 9: Why are you plotting sedimentary rocks on this diagram (i.e. the black plus signs)? Maybe I'm missing something but I don't understand why plotting sedimentary rocks on this diagram is useful.

Line 423: By definition lavas are not intrusive therefore you erupt and not intrude a lava.

Line 438: Not the discrimination diagrams themselves, rather the use of immobile element ratios.

Line 452-453: rhyolites A-X does not mean anything to the reader, simply state that they are from different rhyolite units.

Line 468: List some examples of common mobile elements.

Line 480: Are you saying albitisation, chlorite and sericite are produced during distal seafloor alteration? Chlorite is a high T mineral (>300°C), ambient seafloor alteration typically forms smectite with Fe-oxides and maybe minor zeolite minerals. Please clarify if this is seafloor alteration or if its related to later metamorphism.

Line 482: little → minor

Figure 10: As per my previous comment, please include a key in each figure. Why not add fields indicating typical proximal or distal facies or arrows from proximal to distal in AI-CCP space?

Line 512: higher → upper portion

Line 526: $\Delta C_i$ – what is this? You do this in the next line but the explanation should come first.

Figure11: I think these type of drillcore log figures should be included earlier along with core images for context. Units missing on Y axis. What are the little black lines with arrows? Faults presumably – please state.

Line 548: Why are they deep fluids? Why not just high temperature?

Line 556: How enriched? Give a value.

Line 577: Replace either broadly or broader, repetition.

Line 586: Saying "later stage" is misleading as this still formed on the seafloor – in fact from active SMS deposits some studies suggest the addition of Si very early (You and Bickle, 1998, Nature).

Line 590: delete "characteristic"

Line 618: Example of elements that are used is needed.

Line 619: Define behavior of the element? Presumably you are referring to its solubility which is controlled by temperature, Fo2 and pH.

Line 622: I'm not sure this statement is correct – there are many issues associated with Tl and Hg, especially when using whole rock digestion methods as they are highly volatile. Moreover, Sb generally occurs at very low concentrations.

Line 659: Geochemically anomalous envelope of what? State which elements are anomalous.

Line 682: But why are Sb and Tl below detection? Something to do with temperature perhaps?

Line 701: Or that these elements were not present in the first place.

Line 713: I would present argument 1 and 2 and then discuss why you favour one over the other, in other words, remove the section "in which case…"

Line 729: This section is unclear to me, I'm left wondering what argument you favour? Please clarify.

Line 731: Indicative trace elements? Which trace elements and indicative of what?

Line 734: Delete into the sea bottom → exhaled at the seafloor

Line 735: I'm sorry, I don't understand what is being said here. You don't see an alteration halo above the deposit – but I thought there was a fault zone (later) with displacement so why would you expect to see any halo?

Line 741: Please specify what these indicator elements are, the reader should not have to go back and check.

Line 749: Define proximal, you say 500m in the next line, add it in brackets here.

Line 757: Why do they have to be volatile elements? Why not just elements?

768: Geochemical alteration halos in… VMS deposits in the IPB fall…

Line 771: Define "high transportation temperatures" – generally >280°C

Line 784/785: Delete this sentence, you do not discuss Rb or Sr so don't include this.

Line 788: And lower detection limits.

Line 794: Hollis et al., 2021 in Minerals is also a good VMS reference to add.

Lin 806-811: This describes the figure and should be re-located to the figure caption.

Line 820: But aren't you analysing powders that *should* be homogenous? Nice point to show that you need X amount of analysis to replicate bulk sample concentrations.

Line 826: But surely the grain size of different mineral is far more important? For example analysing a granite would be no good as the individual minerals are very coarse vs. a shale which is fine grained.

Line 830:  Unreasonably high values → unreasonably high concentrations relative to whole rock values.

Line 854: Is Myra Falls a VMS deposit? Please specify.

Figure 14: What do the grey dashed boxes represent? In the caption you state "laboratory", what method does thus follow? Is it ICP or desktop XRF.

Figure 15: Again specify what is meant by laboratory – I think these plots are very useful.

Line 876: Define what you mean when you say "nugget effect" I would not think the nugget effect applies to an element such as Zn, are you referring to the analyses of mineral inclusions?

Line 901: Discriminant → discrimination

Line 944: The biggest issue I see with this statement is that there is no consideration given to mineral grain size – i.e. a coarse vs fine grained rock. I think this would have a profound effect on the data collected, and, as you demonstrate, different elements partition into different minerals, this would cause a serious bias in the results.

In general I think this study should be simplified and the discussion sub-divided into clear topics, maybe, proximal and distal alteration at first and then afterwards into sericite, albite, chlorite (etc.). You could then go on to compare with other VMS deposits. This would make it easier to follow where you are in relation to the ore deposit. The addition of representative drill core samples is essential and will provide much needed complex. The title of the manuscript mentions the use of pXRF but this is currently underrepresented in the text, including some of the figures that are currently in the supplementary information would strengthen this aspect in the manuscript.

Supplementary material: There is a lot of information included in the supplementary material, some of which (such as the logs) could be useful to include in the main text. I also think it would be useful to include the figures such as S3 (section S1.3) in the main text, I think this is great data and is very important when considering the application of pXRF data in industry.

---

## Referee Comment (RC2)

Review of:

**Vectors to ore in replacive VMS deposits of the northern Iberian Pyrite Belt: mineral zoning, whole rock geochemistry, and use of portable XRF**

By:

Guillem Gisbert, Fernando Tornos, Emma Losantos, Juan Manuel Pons, Juan Carlos Videira

The submitted manuscript presents a study aimed at identifying geochemical indicators of VMS deposits, applying geochemical and petrographical data to the Aguas Teñidas deposit of the Iberian Pyrite Belt as a case study. The authors discuss in considerable detail the implications of their data for economic exploration in similar settings worldwide. Additionally, the latter portion of the study focusses on a rigorous test of the applicability of handheld XRF to identify the described geochemical trends, comparing directly data derived in the field and data derived from conventional laboratory analysis (WD-XRF/ICP-OES/ICP-MS).

The manuscript is well written, the background material is, to the best of my knowledge, adequately cited and provides sufficient context, the figures are clear and well-designed, and the supplementary information provides a wealth of information which is highly relevant to the results of the study. I congratulate the authors on a well presented, detailed, and thorough manuscript which I enjoyed reading.

I believe that the results of the study are highly relevant and entirely worthy of publication. However, my only major concern lies in the overall clarity of the manuscript. The manuscript is approximately 16,000 words in length, and some of the main points that the authors wish to disseminate are a little lost in the structure of the paper, which can at times be quite confusing, particularly when considering both the volume and the diversity of the presented data. Sometimes it is hard to tell where the literature data stop and the new data presented in this study begins. I think the structure could be revised to compensate for this without necessitating the removal of material. For example, I would suggest that the section currently entitled "4. Results and discussion" should be divided to ensure that the results of the study are presented in a strictly non-interpreted manner, followed by a discussion that refers back to the previously presented data. The new discussion section, now unburdened from the voluminous description of the data, could then be more thematic in its approach, discussing key topics and referring back to the results. This decoupling of the interpretive and non-interpretive aspects of the study would, I believe, make it significantly more impactful and less confusing for the reader.

Finally, I think the pXRF component of the manuscript is a little understated, and appears as something of an 'add-on' at the end of the manuscript. This is a bit of a shame, as I feel this component of the study is extremely valuable. To some extent, this might be the result of having confined so much of the pXRF details to the supplementary information, although I appreciate there may be manuscript length constraints, which I leave to the discretion of the editor. Nevertheless, I think there is still some scope to tie this component of the study in a little more than it is currently. I think this is one of the best aspects of the paper and a valuable contribution to ongoing studies surrounding hand held XRF.

Below, I provide some more specific comments on the text:

Line 3: Worth spelling XRF out in full in the title?

Line 14: This sentence is quite long and hard to follow

Line 17: VMS districts where? Worldwide?

Line 17-21: Worth numbering each clause for clarity I think.

Line 21: First instance of XRF – could be defined here.

Line 22: hydrothermal alteration *zone*?

Line 25: Indices or indexes? Perhaps both are valid.

Line 26-27: The first clause is more of a statement of quantity (enrichment), second clause is a process (leaching). This reads a little strangely to me, but it might just be me.

Line 40: Change 'and an inevitable' to 'as well as an inevitable'?

Line 43: the mineral systems of this study?

Line 60: This figure is extremely helpful. If anything, I would make it larger. Also, the term CCPI is not defined in the caption.

Line: 92: Change 'ambient rocks' to 'local lithologies'?

Line 97: 'de centre' should presumably be 'the centre'?

Line 102: geochemical zoning patterns?

Line 108: You can probably delete 'related to' in both instances

Line 119: I would give the ages here for context

Line 135: Could clarify here, are these silicic sediments presumably?

Line 137: The end of this paragraph could be numbered to enhance clarity

Line 143: Change 'Subsequent to deposition' to ' Following deposition'

Line 148: Greenschist should be one word

Line 151: This paragraph is quite disjointed and hard to read.

Line 156: As a native English speaker, I instinctively want to change this to 'The Aguas Teñidas deposit', both here and at various points throughout the text. However, I defer entirely to the authors knowledge on the correct usage of the name.

Line 175: This figure could use some more information. Is the orientation given on the upper image the same for both? Is the vertical scale the same as the horizontal scale? The numbered black lines are boreholes, but this is more apparent coming back to the image than when first seeing it.

Line 183: Change 'At a large, deposit, scale' to 'At deposit scale'?

Line 205: MATSA is given in the caption but not defined.

Line 220: No need to abbreviate Unit I think. Also, the first lithology in the table should contain fewer felsics dykes rather than less.

Line 227: Greenschist facies was mentioned earlier. Was that not the IPB?

Line: 238: You could give more info on the other techniques used. E.g. SEM and petrographic microscope? Also, I would give more specific info on the sample prep and the techniques used for the

geochemistry. I think this is critical for the comparison between conventional lab and hand held XRF. What sort of precision and accuracy were obtained from the conventional techniques? How does this compare?

Line 245: I think the classification of cores and being proximal or distal could be made much more apparent. It is a little hard to work out which cores are which at the moment. Even a very small table to refer to would be useful.

Line 257: This supplementary info is quite important, and could probably be more included in the main body of the manuscript than it is currently. Even some of the key figures could be helpful to make the points.

Line 266: This extra sentence doesn't need to be on its own.

Line 273: Change 'allows extending them' to allows their extension'

Line 275: Given the range of alteration events that the IPB has undergone, I think a brief summary table/figure would be justified, so that the reader can immediately establish which parts of the deposits were subjected to which sorts of alteration and when.

Line 278: Change 'materials' to 'lithologies'?

Line 283: Change 'crystals' to 'phases' and give examples? Olivine and pyroxene presumably.

Line 288: Change 'with alteration degree' to 'with the degree of alteration'. Also, the following sentence is a little unclear. I think it might make sense if you just removed the 's' from 'phenocrysts', depending on what you are saying is a pseudomorph of what.

Line 294: Change 'presenting' to 'containing'

Line 299: Hydrothermal alteration *zone*?

Line 311: These are presumably sample numbers as well as depths? Worth adding the units for clarity?

Line 347: Should this be 'interface'?

Line 348: Could change 'disseminations' to 'disseminated bodies'?

Line 356: Clarify where Bathhurst Mining camp is.

Line 358: Change 'halos' to 'haloes'

Line 371: '*and* 3) the trace...'

Line 380: The two plots should be labelled as A and B

Line 385: Change 'recognizing' to 'the recognition of'

Line 423: Remove intruded. Intrusive lavas could cause some confusion.

Line 425: Change 'drillings' to 'drill cores'?

Line 450: Change 'similarly' to 'similar'

Line: 483: Change 'from crustal origin' to 'of crustal origin'

Line 490: I would probably add a key to each of the figures. There is enough space I think, and it makes life a lot easier for the reader.

Line 508-512: A valuable point, well made.

Line 526: This is the first occurrence of this delta value. I appreciate it is defined afterwards, but it should ideally be defined at first occurrence to avoid confusion.

Line 540: These are excellent summary plots, I feel that a few more of these would be both beneficial and entirely justified.

Line 557: These depletions could be quantified for context.

Line 568: Change 'to chlorite-rich' to 'with chlorite-rich'

Line 574: This sentence could be integrated with the paragraph below.

Line 577: Two occurrences of 'broad'

Line 580: Here and elsewhere, these appears to be inconsistent use of elements reported as either the oxide of the element. Is this for a reason? If not, it should be consistently applied.

Line 633: Change 'Consistently to' to 'In agreement with'

Line 650ish: This where I feel another summary figure which highlights some of the geochemical trends described in the text would be useful. I'm not sure what is possible, but perhaps something that puts these trends into perspective in the context of the deposit would be useful.

Line 662: Change 'Differently from' to 'In contrast to'

Line 707: Wirth adding a reference or two for this statement on percolation.

Line 733: ',we searched *for* geochemically'?

Line 765: Spell out mercury at beginning of the sentence

Line 768: Worth clarifying where the Rosebery deposit it?

Line 815: This key could be a little more clear. I found it hard to establish what was what in the plot.

Line 819: Is this still talking about pressed pellets? If so, why is homogeneity a problem?

Line 839: Change 'variations within the mantle…..' to 'between mantle…'

Line 840: Change 'or' to 'and'

Line 850: Change 'on its own' to 'on their own'

Line 853: At this point, a significant portion of the major element section is spend discussing trace elements.

Line 869: This is a very valuable figure. This sort of comparison would be worth adding to, if you have more data.

Line 882: Change 'too high' to 'unacceptably high'

Line 901: 'Discrimination' diagrams

Line 902: change 'elaborated' to 'presented'

Line 912: Change 'alkalis' to 'alkali'

Line 913: Correct tense? Was released?

Line 947: 'In addition, *the presented* data…'

---

## Editor Comment (EC1)

To Guillem Gisbert et al.,

Thank you for your submission "Vectors to ore in replacive VMS deposits of the northern Iberian Pyrite Belt: mineral zoning, whole rock geochemistry, and use of portable XRF" to the Solid Earth special issue for State of the art in mineral exploration. The manuscript has been reviewed by two experts in great detail. Overall, the reviewers emphasise the importance of the study for mineral exploration, praise the scientific merit and general story of the contribution. Both reviewers also note areas of improvement that can be made to the writing style and structure of the presentation, and suggest areas of improvement relating to the use of supplementary material, pXRF data and core photographs. I agree with the comments made by the reviewers, and suggest minor corrections can be made to the layout of the paper to improve it's clarification for a wider audience.

Both reviewers have provided general feedback and line-by-line corrections which should bring the written quality up to a higher standard. Both have also suggested ways to improve the layout of the paper, which I will leave to the authors to decide which and how to follow (either by focusing on one particular suggestion or integrating the ideas of both reviewers to improve the flow of the manuscript). Shortening in places and a clearer breakdown of headings and subheadings for the results and discussion will go a long way to addressing these concerns. It is also noted that the pXRF data could be better utilised within the main text. Although this probably seems contradictory to the "please shorten" narrative, it would be beneficial to bring some of the supplementary pXRF information into the text, such as an additional figure and more integration into the results and discussion.

Thank you again for your submission to Solid Earth and State of the art in mineral exploration, and I look forward to reading the revised version.

Kind regards,

**Dr. Liam Bullock**
Guest Editor for Solid Earth Special issue | State of the art in mineral exploration
University of Oxford

---

## Author Comment (AC1)

**REVISION NOTES**

Vectors to ore in replacive VMS deposits of the northern Iberian Pyrite Belt: mineral zoning, whole rock geochemistry, and application of portable X-ray fluorescence

Guillem Gisbert, Fernando Tornos, Emma Losantos, Juan Manuel Pons, Juan Carlos Videira

MS No.: se-2021-50

Dear Editor,
Please see our responses to the reviewers' comments below. All our comments are written in blue for clarity.

**GENERAL COMMENTS**

We truly appreciate the comments of the reviewers, as these have largely contributed to improve the original manuscript.

Before addressing specific comments from each reviewer, we would like to provide some general comments about some of the main points noted by reviewers.

**1. Structure and writing style**

Both reviewers have expressed concerns on the structure of the paper and on how results, interpretation and discussion are presented to the reader. For instance, Reviewer 1 comments that it is difficult to follow the presentation and discussion of results from different parts of the hydrothermal system; this reviewer suggests providing additional figures and to subdivide information into separate sections for each zone. On the other hand, Reviewer 2 comments that new results, literature data and interpretation are sometimes mixed, and suggests subdividing the manuscript into separate Results and Discussion sections. In addition, both reviewers argue that the manuscript is unclear or confusing in some areas, in part due to the volume and the diversity of the presented data, and that it needs to be simplified.

The main complexity at the time of writing this manuscript was that it presents a large amount of information related to different approaches and techniques (e.g. mineralogy, whole rock geochemistry, use of p-XRF); in addition, to get the most of this information it needs to be presented and described first for different parts of the studied hydrothermal system, then compared between the different parts, interpreted to extract the general picture, and finally compared to other VMS deposits in the IPB and elsewhere. As the reviewers indicate, this jump between different parts of the system and types of data may be confusing for the reader. Addressing the reviewer's comments, we have revised and corrected the manuscript to help the reader follow the presentation and discussion of results. For this purpose, we have modified Figure 3 to clearly indicate what parts of the hydrothermal system we refer to when using expressions such as "distal cores", "distal stockwork", "deep stockwork", etc., and in which drill cores these were sampled. Additionally, we now provide schematic stratigraphic columns for all the main drill cores in which information on the lithology, alteration and mineralization is presented. We have also added titles which subdivide the information into smaller more specific sections dealing with each part of the system or aspect of the discussion,

which help the reader locate themselves within the general presentation and discussion of results. And following the suggestions by Reviewer 1, we have moved all diagrams showing the whole rock geochemistry data along studied cores from the Supplementary materials to the main manuscript.

Regarding the suggestion by Reviewer 2 to produce separate Results and Discussion sections, we think that this would make the manuscript difficult to follow for the reader because there is too much information for the reader to keep in mind and, thus, when reading the discussion section they would have to go back and look at the results which are discussed at each moment. Therefore, we think that it is better to maintain the original structure of the paper, in which, for each type of data, results are presented and then interpreted and discussed before moving into the next type. Nevertheless, addressing the comment by Reviewer 2 we have revised and corrected the manuscript to clearly separate presentation of results and interpretation and discussion within each section. Finally, some parts of the manuscript have been simplified.

**2. Section on p-XRF application to the study of vectors to ore**

Both reviewers argue that the information on the applicability of p-XRF for the use of the vectors to ore described in this manuscript is an important contribution of our work which is, however, under-represented in the main text and mostly relegated to the Supplementary material. Thus, they suggest moving a substantial part of this information from the Supplementary materials to the main text.

We agree with the reviewers that most of the information provided in the Supplementary materials can be highly valuable for researchers or companies wanting to apply p-XRF to the exploration of VMS systems. However, we think that the main focus of the manuscript should be the vectors to ore, and not the technique used to analyse them. Adding a long section on technical aspects related to the use of p-XRF devices could deviate the attention of the reader from the vectors themselves. We also think that the current length of the manuscript, which is already of 20,000 words including references and figure captions, argues against extending it further. Additionally, we think that presenting this information in the Supplementary materials, without length limitations, allows us to provide it in a clearer way. Nevertheless, we have added a short section at the beginning of the p-XRF part of the manuscript in which we stress that the quality of p-XRF measurements can be affected by factors that are assessed and discussed in detail in the Supplementary materials, and briefly comment two of these factors which are mentioned in the subsequent discussion. Thus, now the reader is informed several times along the manuscript that detailed information on technical aspects regarding p-XRF measurements is available in the Supplementary materials, so interested readers should have no problem in accessing it, while readers not interested in the use of p-XRF will not be distracted by technical information in the main text.

**REPLY TO COMMENTS OF REVIEWER 1**

The paper entitled "Vectors to ore in replacive VMS deposits of the northern Iberian Pyrite Belt: mineral zoning, whole rock geochemistry, and use of portable XRF" presents mineralogical and bulk geochemical datasets from the Aguas Tenidas VMS deposit in the Iberian Pyrite Belt. The study focuses on the application of trace metal ratios and

concentrations in bulk-rock powders to the discovery of VMS mineralisation. In addition to this the study compares data obtained from whole-rock (ICP + XRF) data with portable XRF analysis. The latter in my opinion is very important and gives the paper a clear application to the mining and mineral exploration sector. In parts I found the structure confusing and think it needs to be more clearly separated into proximal and distal facies in terms of both mineralogy and geochemistry. I struggled to link the core and core location to the geochemical data. To overcome this, I suggest including representative core photos and more logs like those included in Figure 11. Generally there was a lack of quantification throughout with terms such as "hot", "proximal", "slight" or "laboratory" but no explanation/quantification was given as to what these terms specifically meant. Clearly there is an interesting story here but the manuscript was very long and unclear in some areas. At the moment I think the key points are lost in amongst extraneous information which makes the paper confusing to read.

Please see general comments.

Please see my detailed comments below:

Line 1: Title, consider "replacement style" instead of "replacive"

We have chosen to use "replacive" following Tornos (2006) and Tornos et al. (2015).

Line 3: Do not abbreviate XRF in the title, spell it out.

The title has been changed as suggested.

Line 9: I would not state mineral systems – I think ore deposits or economic mineralisation would be more appropriate. Mineral systems gives the impression of mineralogy or mineral-scale chemistry.

The text has been changed to "types of ore deposits".

Line 10: Define what you mean by "vectors to ore" – geochemical, mineralogical, other or both?

In this case we refer to vectors to ore in general. We feel that specifying is not needed as "vectors to ore" is a commonly used concept.

Line 11: suggest swapping "outstanding" for "world-class"

The text has been changed as suggested.

Line 21: What does XRF stand for?

The full name of the technique is now stated in this first occurrence of the term.

Line 31: I question this statement as the analytical spot size of a pXRF is a few microns (?). What if you have coarse mineral grains? The mineral scale partitioning of different elements between minerals will be important and severely bias your data.

In standard p-XRF equipment spot size is 3 to 8 mm. Still, coarse grainsize can produce significant bias if not enough spots are analysed to characterize a given rock or sample. This is explained in the main text, but we think that this clarification is not needed in the abstract.

General comment, throughout I suggest change "replacive" to replacement type.

We have chosen to use "replacive" following Tornos (2006) and Tornos et al. (2015).

Line 40: Under covering? Are you referring to increased overburden or simply that we are having to hunt in hard to reach places such as tundra and rainforests? Please clarify.

The text has been modified to make the point clearer.

Figure 1: Nice figure, some labels such as CCPI need to be explained. What do the two black dashed lines represent?

The meaning of the CCPI abbreviation is now explained in the figure caption. The two dashed lines delimit the stratigraphic level of the ore; this information has been added to the key.

Line 62: Give an approximate tonnage, see Galley et al., 2007 (I think their figure is >1.5 Bt of sulfide ore).

The figure is given in the geological context, but it is now also given here.

Line 83: What do you mean by mineralogical zoning? Alteration halo?

Yes. The text has been modified.

Line 92: Define "hot" and I suggest replacing with "high temperature" ~350°C

Done.

Line 92: ambient -> surrounding

Done.

Line 93: feeder zone… that directly underlies the massive sulfide mound

Done.

Line 93: You keep mentioning replacement, you should add a sentence explaining exactly what you mean by this and how this differs from the typical exhalative VMS deposit model.

A sentence has been added to the introduction.

Line 103: I would separate this into hypogene mineralizing processes and then anything that is

secondary in origin – why are the secondary processes important? Because they can modify the primary signatures.

An additional sentence has been added to make this point clear.

Line 108: remove "related to"

Done.

Line 116: You state a tonnage for La Zarza why not for Neves Corvo etc.?

Tonnages have been added to the other deposits.

Line 119: Specify an age next to the geological time periods, this would be easier for the reader to understand.

The ages have been added.

Lien 130: oldest to youngest.

Done.

Line 133: Chemical sediments? Please specify a rock type.

It is now specified. These are mainly chert and Fe-Mn-rich sediments.

Line 134: I would insert a reference back to Fig. 2 here as you are discussing the location of the deposit.

Done.

Line 141: very minor point… comma is missing after "et al." on Velasco-Acebes et al. 2019 (and some other reference throughout).

The comma has been added.

Line 148: Is there an approximate temperature range for this? Or at least specify the upper temperature limit if possible. Green schist should be one word.

A maximum temperature of 350 °C has been estimated for the IPB; this is now specified. "Greenschist" has been corrected.

Line 151: Use of "ores", are they actually ores, do they have an economic value?

Some of them were mined.

Lien 151: This should be integrated with above paragraph or deleted – it seems a little out of place.

Important information has been integrated into the previous paragraph, and the rest deleted.

Line 155: "The" Agus Tenidas deposit

Done.

Line 160: Could you state an approximate depth of burial/overburden?

This information is provided in the next sentence, which has been rewritten to make it clearer.

Line 161: Be consistent, is it W or west?

West has been changed to W.

Line 176: (Figure 3) I would instead say aerial view or plan view and cross section view looking (X direction). You should also state what the black lines are, it's obvious to me that they are drill holes (I hope!) but this should be stated.

The figure has been modified, with drill holes added to the legend. "Plan view" is now used in the figure caption, but not "cross section" as the image depicted is not a cross section but how the deposit is seen from the front.

Line 183: Delete "at large"

Done.

Line 190: In what quantity are these minerals present? Presumably trace amounts <1%?

Works describing the mineralogy of the orebody do not quantify their presence; however, "minor amounts" has been added to the text to show that they occur in lower quantities compared to the main sulphides.

Line 193: Avoid 1 sentence paragraphs, integrate with above paragraph.

This sentence has been integrated with the first paragraph.

Line 197: Replace "material" with lithologies.

Done.

Line 201: Dismembered by tectonics? Do you mean faulting during later tectonic events?

Yes. The text has been modified to "subdivided during tectonic deformation"

Line 208: add "The" to the beginning of the sentence.

Done.

Line 213: What do you mean when saying "strong vertical and lateral facies changes"? Please specify, for example, in my mind I think of grain size or rock type, is that correct?

Yes. The sentence has been rephrased.

Table 1: Do not abbreviate unit to U in the table – if you write unit than you can delete the name you repeat in the brackets below and just include the abbreviation – this would make the table clearer.
In the table you use terms like "rich" and "less" can these be quantified?

"Unit" is now not abbreviated. The name in brackets is the local name used by MATSA for each unit (explained in the table caption); it may be the same one or not, and therefore they are written in full.

The terms "rich" and "less" cannot be quantified, as these are generic simplified descriptions provided in Conde and Tornos (2019).

Line 223: Other deposits, such as?

All of them. This is now stated.

Line 227: Earlier in the MS (Line 148) you state up to greenschist facies metamorphism

It is now stated that in the Aguas Teñidas area it only reached the prehnite-pumpellyite facies according to literature.

Line 228: Remobilisation of what?

Metals, it is now stated.

Line 230-231: I would deleted this, you state this is more detail in section 3.1, no need to repeat.

We think that this sentence is good to introduce the reader to the methods section, and to state that the study is based on new samples and not on literature data.

Line 239: Don't start a sentence/paragraph with an abbreviation, spell it out. When you say "whole rock analysis" what does this mean?

The figure is now spelled. We refer to bulk rock analysis. We think that "whole rock" is the most common term to refer to the analysis of the bulk rock, without any kind of separation.

Line 242: Is there a document provided by SGS outlining the detailed methodology? If so, this should be reference here.

Unfortunately, there is not.

Table 2:

MU5 – define "close to MU-2"

In general I found the table hard to follow. This should be moved to the appendix, instead why not include some representative core photographs? I think it would be more useful for the reader to visualise what the core looks like. Core logs for key boreholes as well perhaps.

A figure presenting the columns for all drill cores has been added to provide information on core characteristics in a more graphical way (new Figure 6). In addition, the different areas mentioned in this table and the main text (e.g. distal cores, distal stockwork, central stockwork, etc.) are now marked in Figure 3 to help the reader. We think that this table is more useful in the main text than in the appendix.

Figure 6: Referring back to Fig. 3 for the legend is not appropriate, this needs to be included in this figure.

Figure 6 has been merged with Figure 3.

Line 251: A selection of samples, n =?

The number is now specified.

Line 252: GOLDD Technology? I have no idea what this means, delete or specify.

It's part of the name of the equipment.

Line 266: Avoid 1 sentence paragraphs… integrate with above section.

It now has two sentences as information on where the petrographic study has been performed has been added addressing a comment by Reviewer 2.

Line 273: You mention that it confirms their observations but you need to state what these observations are.

These are now stated.

Line 276: What does distal mean? 10 m or 10 km, this is important and not clear to me.

The different areas (e.g. distal cores, distal stockwork, central stockwork, etc.) are now marked in Figure 3 to help the reader. Additionally, we have stated the distance in the text.

Line 277: But the rocks surrounding the deposit have also been exposed to prehnite-pumpellyite facies metamorphism so they don't only preserve seafloor metasomatism.

Indeed. But, since they have not undergone hydrothermal alteration, it is where more information can be obtained on seafloor metasomatism. It is now specified in the text that the effect of metamorphism likely induced mineralogical changes.

Line 283: Change "mafic crystals" to igneous glass or groundmass.

We think that the origin of this chlorite are mafic phases, not glass or a fine-grained groundmass. A similar interpretation has been made in other VMS districts (e.g. Australia, Large et al. 2001).

Line 287: You say that sericite-quartz alteration dominates? Is this not a function of host rock, i.e., you might have a higher proportion of felsic lithologies in these drillcore samples?

Yes. It is now reminded that the URD unit is felsic in composition.

Line 299: add zone after hydrothermal alteration.

Done.

Line 310: "Minor to rare", this is subjective and should be quantified – <5 vol.%?

A formal count has not been made, but it's approximated at <10%, depending on rock composition.

Line 311 and 313: Specify units for the depth measurements, presumably m.

It is now specified that samples names are their depth along the core in m.

Figure 7: petrographic microscope picture -> photomicrographs
Again add units after measurements (m) – nice images!

These are sample names, not directly a depth measurement.

Line 334: reference to Fig 7H that shows Fe oxides (?)

It is referenced in the previous sentence, where Fe oxides are now mentioned.

To make section 3.1 easier to understand I suggest sub-dividing it into proximal and distal sections and hangingwall and footwall. It becomes confusing when swapping between different drill holes and you easily lose the reader.

We have added titles to separate information from different parts of the system and help the reader follow the discussion.

Line 365: Yes, I agree that hyperspectral imaging is great for exploration but you focus on drillcore samples, how is this method applicable to your data/deposit? I would summarise your key alteration findings here (i.e. the expected spectral response of proximal vs. distal alteration) and relate this to hyperspectral imaging.

It is now specified that these techniques can be applied to both hand specimens and drill cores. We think that describing in detail the effects of mineralogical changes to the signal in hyperspectral devices is too specific and would not help the reader, so we prefer not to add

this information. However, we cite several works that can be used by the reader as a reference.

Line 370: Unit -> rock, also extra e on halos.

In this case unit is used on purpose because it refers to the URD. Both "halos" and "haloes" are correct, we have decided to use "halos" and have checked the text for consistency.

Line 373: I don't think mineralization related lithologies is what you are identifying, rather it is chemical changes that related to specific hydrothermal/mineralising events.

In this case we are referring to specific lithologies to which mineralization can be associated. For example, in the IPB there is an ongoing discussion on whether VMS deposits occur associated to volcanic units with specific characteristics (either composition, age, texture, etc.). The sentence has been rephrased.

Line 374 are "to ore deposits" after pathfinder

Done.

Figure 8: In the text you reference figure 8b – A and B are missing on the figure.

The plots are now labelled.

Figure 9: Why are you plotting sedimentary rocks on this diagram (i.e. the black plus signs)? Maybe I'm missing something but I don't understand why plotting sedimentary rocks on this diagram is useful.

The sedimentary rocks are shown as a reference because there are volcaniclastic rocks that have a siliciclastic component; this is now explained in the text. The key has been added to the figure.

Line 423: By definition lavas are not intrusive therefore you erupt and not intrude a lava.

Correct, the term was used incorrectly. "Lavas" has been changed to "lavas/sills" because in some occasions there is uncertainty on the effusive/intrusive character of these rocks.

Line 438: Not the discrimination diagrams themselves, rather the use of immobile element ratios.

It is the combination of specific ratios in the proposed diagrams. The sentence has been changed.

Line 452-453: rhyolites A-X does not mean anything to the reader, simply state that they are from different rhyolite units.

The names of the rhyolites have been removed.

Line 468: List some examples of common mobile elements.

Done.

Line 480: Are you saying albitisation, chlorite and sericite are produced during distal seafloor alteration? Chlorite is a high T mineral (>300°C), ambient seafloor alteration typically forms smectite with Fe-oxides and maybe minor zeolite minerals. Please clarify if this is seafloor alteration or if its related to later metamorphism.

It is now specified that metamorphism could have produced mineralogical changes, but not significant chemical ones. In the petrography section it is now explained that the current mineral assemblage is the result of alteration processes and subsequent metamorphism. Whereas albitization is most likely related to seafloor metasomatism, as suggested by the reviewer the current chlorite and muscovite may be to some extent the product of metamorphic recrystallization. However, although not explained in this manuscript, the extremely fine grain in shales indicates very minor recrystallization.

Line 482: little -> minor

Done.

Figure 10: As per my previous comment, please include a key in each figure. Why not add fields indicating typical proximal or distal facies or arrows from proximal to distal in AI-CCP space?

The key has been added. Although indicating the fields suggested by the reviewer could be useful, it is not straightforward (even if done only for the URD rocks) because initial seafloor alteration produces a significant dispersion in the whole rock geochemistry data, so trends associated to hydrothermal alteration depart from different points in the diagram. In addition, whereas there is a general trend towards the chl+py vertex when moving towards the centre of the hydrothermal system, there are deviations; for example, rocks in and around the distal stockwork in AE-68 and AE-69 present carbonate alteration, which generates a sub-trend towards the dol/ank compositions. Nevertheless, generic arrows have been added to guide the reader.

Line 512: higher -> upper portion

Done.

Line 526: ΔCi – what is this? You do this in the next line but the explanation should come first.

The order of the sentences has been changed.

Figure11: I think these type of drillcore log figures should be included earlier along with core images for context. Units missing on Y axis. What are the little black lines with arrows? Faults presumably – please state.

The diagrams of all cores with whole rock geochemistry data have been added. Arrows (faults/thrusts) have been added to the key. The unit of the Y axis is now stated in the figure caption.

Line 548: Why are they deep fluids? Why not just high temperature?

Hydrothermal fluids are indeed hotter than surrounding ones. "Deep" has been removed from the sentence.

Line 556: How enriched? Give a value.

Example values are now given for sample DST-332/251.5 in the centre of the deep stockwork.

Line 577: Replace either broadly or broader, repetition.

"Broadly" has been changed to "largely".

Line 586: Saying "later stage" is misleading as this still formed on the seafloor – in fact from active SMS deposits some studies suggest the addition of Si very early (You and Bickle, 1998, Nature).

It is now stated that this is an interpretation by Bobrowicz (1995) and that "later" refers to still during the hydrothermal event.

Line 590: delete "characteristic"

Done.

Line 618: Example of elements that are used is needed.

Example elements are now provided.

Line 619: Define behavior of the element? Presumably you are referring to its solubility which is controlled by temperature, Fo2 and pH.

Solubility and partition coefficients are now given as examples.

Line 622: I'm not sure this statement is correct – there are many issues associated with Tl and Hg, especially when using whole rock digestion methods as they are highly volatile. Moreover, Sb generally occurs at very low concentrations.

Despite issues associated to their analysis, which certainly need to be considered, they are still highly useful due to the extent of the halo they produce and thus are receiving attention.

Line 659: Geochemically anomalous envelope of what? State which elements are anomalous.

The sentence has been rewritten to make this clearer.

Line 682: But why are Sb and Tl below detection? Something to do with temperature perhaps?

Most likely. Solubility of these elements in this part of the system may be so high that only very minor amounts of them are kept in it.

Line 701: Or that these elements were not present in the first place.

Given the late character of this band, which intersects the stockwork, and the characteristics of the well-established geochemical halo around it, we think that it is reasonable to assume that the halo was formed and then later disrupted when this structure formed.

Line 713: I would present argument 1 and 2 and then discuss why you favour one over the other, in other words, remove the section "in which case…"

This part of the manuscript has been rewritten and restructured.

Line 729: This section is unclear to me, I'm left wondering what argument you favour? Please clarify.

This part of the manuscript has been rewritten and restructured.

Line 731: Indicative trace elements? Which trace elements and indicative of what?

The elements that we have described that can be used for vectoring VMS deposits. These are now mentioned in the text.

Line 734: Delete into the sea bottom -> exhaled at the seafloor

Done.

Line 735: I'm sorry, I don't understand what is being said here. You don't see an alteration halo above the deposit – but I thought there was a fault zone (later) with displacement so why would you expect to see any halo?

We see an alteration halo above the Aguas Teñidas deposit despite the thrust that separates the massive sulphides from the hanging wall. We checked stratigraphic levels above the host dome away from the mineralization to see if we could see any indication of the influence of hydrothermal fluids which could have been released into the contemporary seafloor.

Line 741: Please specify what these indicator elements are, the reader should not have to go back and check.

Done.

Line 749: Define proximal, you say 500m in the next line, add it in brackets here.

Done.

Line 757: Why do they have to be volatile elements? Why not just elements?

In this case Tl, Sb, and Hg are the most investigated ones because they produce the larger anomalies.

768: Geochemical alteration halos in… VMS deposits in the IPB fall…

Done.

Line 771: Define "high transportation temperatures" – generally >280°C

Done.

Line 784/785: Delete this sentence, you do not discuss Rb or Sr so don't include this.

We think that this information can be useful for the reader. In addition, this ratio is discussed later in the p-XRF section.

Line 788: And lower detection limits.

This information has been included.

Line 794: Hollis et al., 2021 in Minerals is also a good VMS reference to add.

This reference has been added.

Lin 806-811: This describes the figure and should be re-located to the figure caption.

Sentences describing the figure have been moved to the figure caption.

Line 820: But aren't you analysing powders that should be homogenous? Nice point to show that you need X amount of analysis to replicate bulk sample concentrations.

At this point we were referring to hand specimens, but it was not clear. It is now stated.

Line 826: But surely the grain size of different mineral is far more important? For example analysing a granite would be no good as the individual minerals are very coarse vs. a shale which is fine grained.

Grainsize is indeed important. This is the reason why studies aimed at establishing the amount of analyses that need to be averaged to obtain a reliable average composition are important.

Line 830: Unreasonably high values -> unreasonably high concentrations relative to whole rock values.

Done.

Line 854: Is Myra Falls a VMS deposit? Please specify.

It is now specified.

Figure 14: What do the grey dashed boxes represent? In the caption you state "laboratory", what method does thus follow? Is it ICP or desktop XRF.

A key has been added. The meaning of "laboratory" is now specified at the beginning of the section and in the key.

Figure 15: Again specify what is meant by laboratory – I think these plots are very useful.

Please see previous comment.

Line 876: Define what you mean when you say "nugget effect" I would not think the nugget effect applies to an element such as Zn, are you referring to the analyses of mineral inclusions?

We have replaced "nugget effect" for "heterogeneous distribution", as these samples are not mineralized and therefore "nugget effect" is not the most appropriate term.

Line 901: Discriminant -> discrimination

Done.

Line 944: The biggest issue I see with this statement is that there is no consideration given to mineral grain size – i.e. a coarse vs fine grained rock. I think this would have a profound effect on the data collected, and, as you demonstrate, different elements partition into different minerals, this would cause a serious bias in the results.

This issue is described in the Supplementary materials, where we calculate the amount of analysis which are required for each sample in order to obtain reliable average compositions. This is largely controlled by grainsize.

In general I think this study should be simplified and the discussion sub-divided into clear topics, maybe, proximal and distal alteration at first and then afterwards into sericite, albite, chlorite (etc.). You could then go on to compare with other VMS deposits. This would make it easier to follow where you are in relation to the ore deposit. The addition of representative drill core samples is essential and will provide much needed complex. The title of the manuscript mentions the use of pXRF but this is currently underrepresented in the text, including some of the figures that are currently in the supplementary information would strengthen this aspect in the manuscript.

Please see general comments.

Supplementary material: There is a lot of information included in the supplementary material, some of which (such as the logs) could be useful to include in the main text. I also think it would be useful to include the figures such as S3 (section S1.3) in the main text, I think this is great data and is very important when considering the application of pXRF data in industry.

**REPLY TO COMMENTS OF REVIEWER 2**

The submitted manuscript presents a study aimed at identifying geochemical indicators of VMS deposits, applying geochemical and petrographical data to the Aguas Teñidas deposit of the Iberian Pyrite Belt as a case study. The authors discuss in considerable detail the implications of their data for economic exploration in similar settings worldwide. Additionally, the latter portion of the study focusses on a rigorous test of the applicability of handheld XRF to identify the described geochemical trends, comparing directly data derived in the field and data derived from conventional laboratory analysis (WD-XRF/ICP-OES/ICP-MS).

The manuscript is well written, the background material is, to the best of my knowledge, adequately cited and provides sufficient context, the figures are clear and well-designed, and the supplementary information provides a wealth of information which is highly relevant to the results of the study. I congratulate the authors on a well presented, detailed, and thorough manuscript which I enjoyed reading.

I believe that the results of the study are highly relevant and entirely worthy of publication. However, my only major concern lies in the overall clarity of the manuscript. The manuscript is approximately 16,000 words in length, and some of the main points that the authors wish to disseminate are a little lost in the structure of the paper, which can at times be quite confusing, particularly when considering both the volume and the diversity of the presented data. Sometimes it is hard to tell where the literature data stop and the new data presented in this study begins. I think the structure could be revised to compensate for this without necessitating the removal of material. For example, I would suggest that the section currently entitled "4. Results and discussion" should be divided to ensure that the results of the study are presented in a strictly non-interpreted manner, followed by a discussion that refers back to the previously presented data. The new discussion section, now unburdened from the voluminous description of the data, could then be more thematic in its approach, discussing key topics and referring back to the results. This decoupling of the interpretive and non-interpretive aspects of the study would, I believe, make it significantly more impactful and less confusing for the reader.

Finally, I think the pXRF component of the manuscript is a little understated, and appears as something of an 'add-on' at the end of the manuscript. This is a bit of a shame, as I feel this component of the study is extremely valuable. To some extent, this might be the result of having confined so much of the pXRF details to the supplementary information, although I appreciate there may be manuscript length constraints, which I leave to the discretion of the editor. Nevertheless, I think there is still some scope to tie this component of the study in a little more than it is currently. I think this is one of the best aspects of the paper and a valuable contribution to ongoing studies surrounding hand held XRF.

Below, I provide some more specific comments on the text:

Line 3: Worth spelling XRF out in full in the title?

The title has been changed as suggested.

Line 14: This sentence is quite long and hard to follow

The sentence has been simplified.

Line 17: VMS districts where? Worldwide?

Mainly Canada and Australia. We think that specifying is not needed in this case.

Line 17-21: Worth numbering each clause for clarity I think.

Clauses have been numbered. It helps.

Line 21: First instance of XRF – could be defined here.

The full name of the technique is now stated in this first occurrence of the term.

Line 22: hydrothermal alteration zone?

"Zone" has been added to the text.

Line 25: Indices or indexes? Perhaps both are valid.

Both are valid.

Line 26-27: The first clause is more of a statement of quantity (enrichment), second clause is a process (leaching). This reads a little strangely to me, but it might just be me.

Enrichment was used in the text as a process (in this case these elements are enriched through addition from an external source and leaching of other elements). The sentence has been rewritten.

Line 40: Change 'and an inevitable' to 'as well as an inevitable'?

The text has been modified as suggested.

Line 43: the mineral systems of this study?

Yes. "VMS" has been added to the text.

Line 60: This figure is extremely helpful. If anything, I would make it larger. Also, the term CCPI is not defined in the caption.

The meaning of the CCPI abbreviation is now explained in the figure caption. We think that in its current size the figure can be well read and, thus, that making it larger would take unnecessary space.

Line: 92: Change 'ambient rocks' to 'local lithologies'?

The text has been changed to "surrounding rocks" following suggestion by Reviewer 1.

Line 97: 'de centre' should presumably be 'the centre'?

Yes. Thank you.

Line 102: geochemical zoning patterns?

The text has been modified to "geochemical zoning patterns".

Line 108: You can probably delete 'related to' in both instances

Done.

Line 119: I would give the ages here for context

The ages have been added.

Line 135: Could clarify here, are these silicic sediments presumably?

It is now specified that these are silicic.

Line 137: The end of this paragraph could be numbered to enhance clarity

It has been numbered.

Line 143: Change 'Subsequent to deposition' to ' Following deposition'

Done.

Line 148: Greenschist should be one word

The text has been corrected.

Line 151: This paragraph is quite disjointed and hard to read.

The text has been modified and integrated into the previous paragraph following a suggestion from Reviewer 1.

Line 156: As a native English speaker, I instinctively want to change this to 'The Aguas Teñidas deposit', both here and at various points throughout the text. However, I defer entirely to the authors knowledge on the correct usage of the name.

In this case the suggestion has not been followed to avoid repetition of "deposit". However, the manuscript has been modified following this suggestion.

Line 175: This figure could use some more information. Is the orientation given on the upper image the same for both? Is the vertical scale the same as the horizontal scale? The numbered black lines are boreholes, but this is more apparent coming back to the image than when first seeing it.

The orientation is now also given in the lower image, and in the figure caption we state that there is no vertical exaggeration. Drill holes have been added to the legend.

Line 183: Change 'At a large, deposit, scale' to 'At deposit scale'?

Done.

Line 205: MATSA is given in the caption but not defined.

MATSA is now defined in the first figure caption in which it appears, in Figure 3.

Line 220: No need to abbreviate Unit I think. Also, the first lithology in the table should contain fewer felsics dykes rather than less.

"Unit" is now not abbreviated. "Less" has been changed for "fewer".

Line 227: Greenschist facies was mentioned earlier. Was that not the IPB?

It is now stated that in the Aguas Teñidas area metamorphism only reached the prehnite-pumpellyite facies according to literature.

Line: 238: You could give more info on the other techniques used. E.g. SEM and petrographic microscope? Also, I would give more specific info on the sample prep and the techniques used for the geochemistry. I think this is critical for the comparison between conventional lab and hand held XRF. What sort of precision and accuracy were obtained from the conventional techniques? How does this compare?

Information on the optical microscope and SEM is now provided. Reference material measurements provided in the reports by SGS are provided in the Supplementary material 2 to show the precision and accuracy of the method, although the insufficient number of analyses of each reference material hamper adequate calculation of these parameters.

Line 245: I think the classification of cores and being proximal or distal could be made much more apparent. It is a little hard to work out which cores are which at the moment. Even a very small table to refer to would be useful.

The different areas (e.g. distal cores, distal stockwork, central stockwork, etc.) are now marked in Figure 3 to help the reader.

Line 257: This supplementary info is quite important, and could probably be more included in the main body of the manuscript than it is currently. Even some of the key figures could be helpful to make the points.

We agree with the reviewer that this information is important. However, we feel that adding it to the main text would make the manuscript too long and distract the reader from the main point, which is the description of the vectors to ore. Please see general comment 2.

Line 266: This extra sentence doesn't need to be on its own.

Information on where the petrographic study has been performed has been included in this paragraph addressing comment in line 238.

Line 273: Change 'allows extending them' to allows their extension'

Done.

Line 275: Given the range of alteration events that the IPB has undergone, I think a brief summary table/figure would be justified, so that the reader can immediately establish which parts of the deposits were subjected to which sorts of alteration and when.

The text has been modified to help the reader; we think that this is now clear enough so that no table or figure are needed.

Line 278: Change 'materials' to 'lithologies'?

Done.

Line 283: Change 'crystals' to 'phases' and give examples? Olivine and pyroxene presumably.

Done.

Line 288: Change 'with alteration degree' to 'with the degree of alteration'. Also, the following sentence is a little unclear. I think it might make sense if you just removed the 's' from 'phenocrysts', depending on what you are saying is a pseudomorph of what.

Done.

Line 294: Change 'presenting' to 'containing'

Done.

Line 299: Hydrothermal alteration zone?

"Zone" has been added.

Line 311: These are presumably sample numbers as well as depths? Worth adding the units for clarity?

It is now specified that samples names are their depth along the core in m.

Line 347: Should this be 'interface'?

Yes. It has been changed.

Line 348: Could change 'disseminations' to 'disseminated bodies'?

We think that the term "disseminated bodies" would lead the reader to think about macroscopic bodies, when in fact it occurs as fine-grained patches which are mostly visible under the microscope.

Line 356: Clarify where Bathhurst Mining camp is.

Done.

Line 358: Change 'halos' to 'haloes'

Both "halos" and "haloes" are correct. The text has been checked and all "haloes" changed to "halos" for consistency.

Line 371: 'and 3) the trace…'

Done.

Line 380: The two plots should be labelled as A and B

The plots are now labelled.

Line 385: Change 'recognizing' to 'the recognition of'

Done.

Line 423: Remove intruded. Intrusive lavas could cause some confusion.

"Lavas" has been changed to "lavas/sills" because in some occasions there is uncertainty on the effusive/intrusive character of these rocks.

Line 425: Change 'drillings' to 'drill cores'?

We think that drilling is more correct, because it was the drilling that intersected the rock, which was sampled in the drill core.

Line 450: Change 'similarly' to 'similar'

Done.

Line: 483: Change 'from crustal origin' to 'of crustal origin'

Done.

Line 490: I would probably add a key to each of the figures. There is enough space I think, and it makes life a lot easier for the reader.

The key has been added to all the figures.

Line 508-512: A valuable point, well made.

Thank you.

Line 526: This is the first occurrence of this delta value. I appreciate it is defined afterwards, but it should ideally be defined at first occurrence to avoid confusion.

The order of the sentences has been changed.

Line 540: These are excellent summary plots, I feel that a few more of these would be both beneficial and entirely justified.

All diagrams are now provided in the main text.

Line 557: These depletions could be quantified for context.

Example values are given for sample DST-332/251.5 in the centre of the deep stockwork.

Line 568: Change 'to chlorite-rich' to 'with chlorite-rich'

Done.

Line 574: This sentence could be integrated with the paragraph below.

Done.

Line 577: Two occurrences of 'broad'

"Broadly" has been changed to "largely".

Line 580: Here and elsewhere, these appears to be inconsistent use of elements reported as either the oxide of the element. Is this for a reason? If not, it should be consistently applied.

In this case elements were used because this is the form in which changes were reported in Bobrowicz (1995). These have been changed to oxides for consistency.

Line 633: Change 'Consistently to' to 'In agreement with'

Done.

Line 650ish: This where I feel another summary figure which highlights some of the geochemical trends described in the text would be useful. I'm not sure what is possible, but perhaps something that puts these trends into perspective in the context of the deposit would be useful.

A summary figure is provided in the Summary and conclusions section. We think that it is more useful there.

Line 662: Change 'Differently from' to 'In contrast to'

Done.

Line 707: Wirth adding a reference or two for this statement on percolation.

This part of the manuscript has been rewritten and restructured.

Line 733: ',we searched for geochemically'?

Done.

Line 765: Spell out mercury at beginning of the sentence

Done.

Line 768: Worth clarifying where the Rosebery deposit it?

Done.

Line 815: This key could be a little more clear. I found it hard to establish what was what in the plot.

The key and diagram have been modified to make them clearer.

Line 819: Is this still talking about pressed pellets? If so, why is homogeneity a problem?

At this point we were referring to hand specimens, but it was not clear. It is now stated.

Line 839: Change 'variations within the mantle…..' to 'between mantle…'

We were referring to variations within the mantle series, not between the mantle and mixing series. This has now been made clearer.

Line 840: Change 'or' to 'and'

Please see previous comment.

Line 850: Change 'on its own' to 'on their own'

Done.

Line 853: At this point, a significant portion of the major element section is spend discussing trace elements.

It is, but only because we discuss that since major elements may not be used, other elements should be used instead.

Line 869: This is a very valuable figure. This sort of comparison would be worth adding to, if you have more data.

Unfortunately, we don't have more data at the moment.

Line 882: Change 'too high' to 'unacceptably high'

Done.

Line 901: 'Discrimination' diagrams

Done.

Line 902: change 'elaborated' to 'presented'

Done.

Line 912: Change 'alkalis' to 'alkali'

Done.

Line 913: Correct tense? Was released?

Done.

Line 947: 'In addition, the presented data…'

Done.